# Photosensing and quorum sensing are integrated to control *Pseudomonas aeruginosa* collective behaviors

**Sampriti Mukherjee**[1], **Matthew Jemielita**[1], **Vasiliki Stergioula**[1], **Mikhail Tikhonov**[2], **Bonnie L. Bassler**[1,3]*

1 Department of Molecular Biology, Princeton University, Princeton, New Jersey, United States of America,
2 Physics Department, Washington University in St. Louis, St. Louis, Missouri, United States of America,
3 Howard Hughes Medical Institute, Chevy Chase, Maryland, United States of America

* bbassler@princeton.edu

**Data Availability Statement:** All relevant data are within the paper and its Supporting Information files.

## Abstract

Bacteria convert changes in sensory inputs into alterations in gene expression, behavior, and lifestyles. A common lifestyle choice that bacteria make is whether to exhibit individual behavior and exist in the free-living planktonic state or to engage in collective behavior and form sessile communities called biofilms. Transitions between individual and collective behaviors are controlled by the chemical cell-to-cell communication process called quorum sensing. Here, we show that quorum sensing represses *Pseudomonas aeruginosa* biofilm formation and virulence by activating expression of genes encoding the KinB–AlgB two-component system (TCS). Phospho-AlgB represses biofilm and virulence genes, while KinB dephosphorylates and thereby inactivates AlgB. We discover that the photoreceptor BphP is the kinase that, in response to light, phosphorylates and activates AlgB. Indeed, exposing *P. aeruginosa* to light represses biofilm formation and virulence gene expression. To our knowledge, *P. aeruginosa* was not previously known to detect and respond to light. The KinB–AlgB–BphP module is present in all pseudomonads, and we demonstrate that AlgB is the partner response regulator for BphP in diverse bacterial phyla. We propose that in the KinB–AlgB–BphP system, AlgB functions as the node at which varied sensory information is integrated. This network architecture provides a mechanism enabling bacteria to integrate at least two different sensory inputs, quorum sensing (via RhlR-driven activation of *algB*) and light (via BphP–AlgB), into the control of collective behaviors. This study sets the stage for light-mediated control of *P. aeruginosa* infectivity.

## Introduction

Bacterial responses to self-generated and exogenous stimuli influence their survival, persistence in particular niches, and lifestyle transitions such as alterations between being free-swimming or existing as a member of a biofilm. Biofilms are three-dimensional structured

**Funding:** This work was supported by the Howard Hughes Medical Institute, NIH Grant 5R37GM065859, and National Science foundation Grant MCB-1713731 to BLB, and NIH Grant 1K99GM129424-01 to SM. The funders had no role in study design, data collection and analysis, decision to publish, or preparation of the manuscript.

**Competing interests:** The authors have declared that no competing interests exist.

**Abbreviations:** AU, arbitrary unit; BLUF, blue-light sensing using flavin; BV, biliverdin; ESKAPE, *Enterococcus faecium*, *Staphylococcus aureus*, *Klebsiella pneumoniae*, *Acinetobacter baumannii*, *Pseudomonas aeruginosa*, and *Enterobacter* spp.; HK, histidine kinase; LB, lysogeny broth; LED, light-emitting diode; LOV, light–oxygen–voltage; PF, Pass-Filter; qRT-PCR, quantitative Reverse Transcriptase-Polymerase Chain Reaction; rpm, rotations per minute; RR, response regulator; SEM, standard error of the mean; SSA, solid-surface–associated; TB, Tryptone broth; TCS, two-component system; WT, wild type.

communities of bacterial cells encased in an extracellular matrix [1,2]. Bacteria living in biofilms exhibit superior resilience to environmental stresses such as antimicrobials and host immune responses [2,3]. Frequently, biofilm formation is governed by intracellular signaling molecules such as cyclic di-guanosine monophosphate [4,5] and extracellular signaling molecules such as quorum-sensing autoinducers [6,7]. Quorum sensing is a cell-to-cell communication process that relies on the production, release, and population-wide detection of autoinducers [8,9]. Quorum sensing allows groups of bacteria to synchronously alter behavior in response to changes in the population density and species composition of the vicinal community. Many pathogenic bacteria, including the global pathogen *P. aeruginosa*, require quorum sensing to establish successful infections [10]. Here, we show that quorum-sensing signaling converges with the detection of and response to another sensory cue, light, to control biofilm formation and virulence factor production in *P. aeruginosa*. We define the pathway connecting the light and quorum-sensing inputs to the virulence and biofilm outputs.

Light is a common environmental cue that is detected by photoreceptors present in all domains of life [11,12]. Particular photoreceptor photosensory domains are activated by specific wavelengths of light [13]. Photoreceptors fall into 7 families depending on the structure of the light-absorbing chromophore: rhodopsins, xanthopsins, cryptochromes, light–oxygen–voltage (LOV)-domain–containing phototropins, blue-light sensing using flavin (BLUF)-domain proteins, cobalamin (Vitamin B12)-binding proteins, and phytochromes [12–15]. In bacteria, the most abundant photoreceptors are phytochromes [16], typically possessing an amino-terminal chromophore-binding domain and a carboxy-terminal histidine kinase (HK) domain. Bacteriophytochromes assemble with the chromophore called biliverdin (BV) [17]. Surprisingly, very few bacteria encode a partner response regulator (RR) in close proximity to the gene specifying the bacteriophytochrome [18], leaving the systems mostly undefined.

In the human pathogen *P. aeruginosa*, LuxR-type quorum-sensing receptors, which function as transcriptional activators when they are bound to their partner autoinducers, are required for virulence and biofilm formation [19,20]. In this study, we examine the mechanism by which the *P. aeruginosa* LuxR-type quorum-sensing receptor called RhlR represses biofilm formation [21,22]. A genetic screen reveals that RhlR activates the expression of the *algB–kinB* operon (Fig 1). KinB and AlgB are a TCS sensor HK and partner RR, respectively [23,24]. KinB is required for acute infection in zebrafish embryos, in which it controls production of virulence factors such as pyocyanin and promotes motility [24,25]. In the context of our screen, we find that phospho-AlgB (AlgB-P) is a repressor of biofilm formation and KinB is a phosphatase that dephosphorylates and thereby inactivates AlgB. Using genetic suppressor analysis and in vitro phosphorylation assays, we discover that BphP is the HK that phosphorylates and activates AlgB, enabling AlgB to repress biofilm formation and genes encoding virulence factors (Fig 1). BphP is a far-red-light–sensing bacteriophytochrome [26], and indeed, we demonstrate that *P. aeruginosa* biofilm formation and virulence gene expression are repressed by far-red light. Phylogenetic analyses show that the KinB–AlgB–BphP module is conserved in all pseudomonads, and moreover, AlgB is present in the majority of bacteria that possess BphP orthologs. This final finding suggests that the BphP–AlgB interaction is widespread. As proof of this notion, we show that *P. aeruginosa* BphP can phosphorylate AlgB orthologs from α-, β-, and γ-Proteobacteria. Thus, we propose that AlgB functions as an integrator that conveys multiple environmental cues, including those specifying population density and the presence or absence of light into the regulation of collective behaviors (Fig 1). We further predict that AlgB functions as the partner RR for BphP in all bacteria that possess BphP as an orphan HK. The downstream signal transduction components and the outputs of

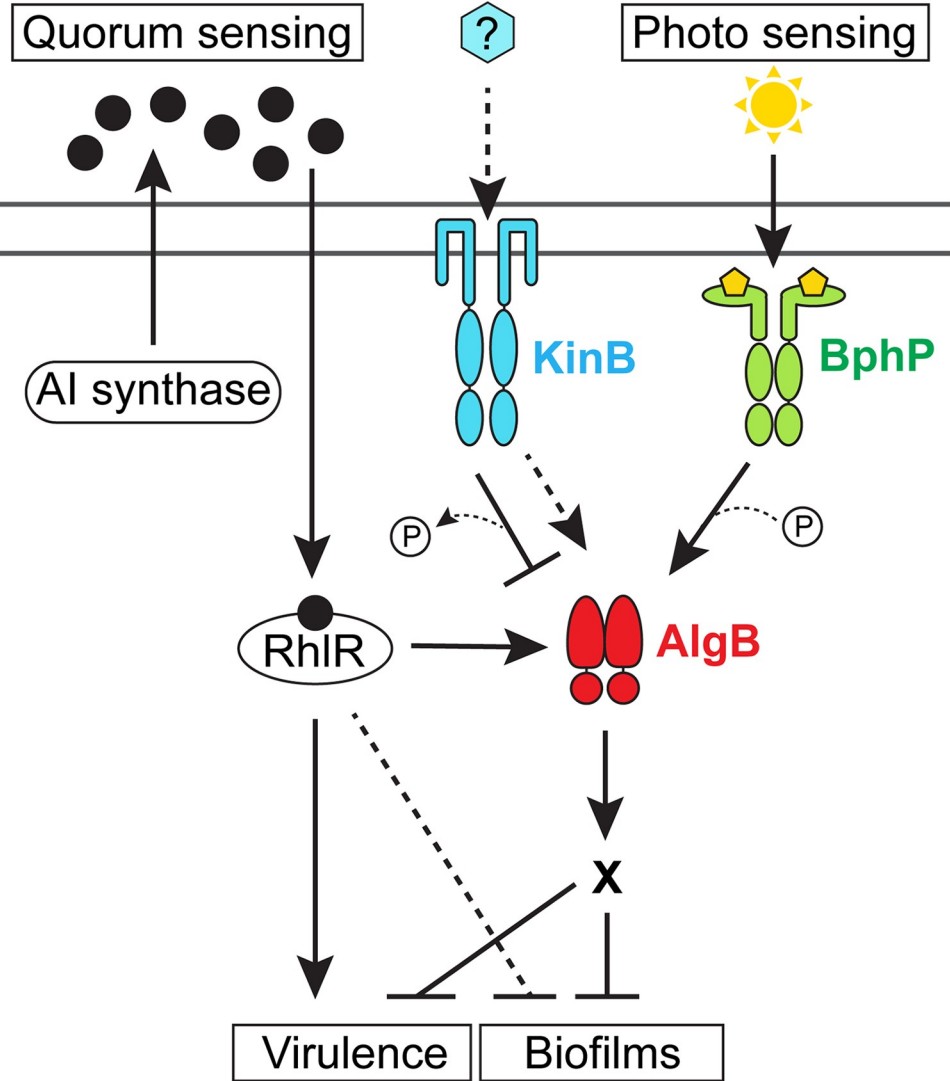

**Fig 1. Model for *P. aeruginosa* integration of quorum-sensing and photosensing information into the control of virulence and biofilm development.** The RhlR quorum-sensing receptor binds its partner AI produced by either the RhlI or PqsE autoinducer synthase (black circles) at high cell density [22]. The RhlR–AI complex represses biofilm formation and virulence gene expression by activating transcription of the *algB–kinB* operon encoding the KinB HK and the AlgB RR, the latter an indirect repressor of biofilm formation. KinB antagonizes AlgB by dephosphorylation. The stimulus (blue hexagon) for KinB is unknown. In the presence of this putative stimulus, the hypothesis is that KinB functions as a kinase for AlgB (dashed arrow). Photosensing stimulates the BphP HK to autophosphorylate and subsequently transfer the phosphoryl group to AlgB to activate AlgB. AlgB-P activates transcription of genes required for repression of group behaviors such as biofilm formation and virulence. A "P" in a circle denotes addition or removal of a phosphate moiety (dotted line). X denotes that the genes functioning downstream of AlgB in the process are not known. The RhlR–AI complex directly activates virulence gene expression and also represses biofilm formation by additional unknown mechanisms (dashed T-bar). AI, autoinducer; HK, histidine kinase; RR, response regulator.

photosensory cascades are not known in the majority of nonphotosynthetic bacteria that possess them, making their physiological roles difficult to discern. This study provides the entire cascade—light as the input, BphP as the detector, AlgB as the signal transducer, and biofilm formation and virulence factor production as the outputs—enabling insight into light-driven control of bacterial behaviors.

## Results

### KinB activates and AlgB represses RhlR-dependent group behaviors

We recently discovered that the *P. aeruginosa* quorum-sensing receptor RhlR represses biofilm formation [21,22]. Specifically, on Congo red agar biofilm medium, wild-type (WT) *P. aeruginosa* UCBPP-PA14 (hereafter called PA14) exhibits a rugose-center/smooth-periphery colony biofilm phenotype, while the Δ*rhlR* mutant forms a larger hyper-rugose colony biofilm (Fig 2A). To determine the mechanism by which RhlR impedes colony biofilm formation, we randomly mutagenized the Δ*rhlR* strain using the transposon Tn5 IS50L derivative IS*lacZ*/*hah* [27] and screened for colonies exhibiting either a WT or a smooth colony biofilm phenotype. Our rationale was that inactivation of a gene encoding a component that functions downstream of RhlR in biofilm formation would sever the connection between RhlR and repression of colony biofilm formation. We screened 5,000 transposon insertion mutants. Strains harboring insertions located in genes encoding hypothetical proteins, proteins involved in twitching motility, and proteins required for Pel polysaccharide synthesis all produced smooth colony biofilms (S1 Table). Most of these genes were already known to play roles in *P. aeruginosa* biofilm formation [28]. Here, we focus on one transposon insertion mutant that exhibited a smooth colony biofilm phenotype that mapped to the gene *PA14_72390* encoding the KinB transmembrane HK (Fig 2A) [23,24,25]. *kinB* is located immediately downstream of *algB* in a dicistron that is conserved in all sequenced pseudomonads (S1A Fig; www.pseudomonas. com). To verify that KinB plays a role in colony biofilm formation, we generated an in-frame markerless deletion of *kinB* in the chromosomes of the WT and Δ*rhlR* strains. Both the Δ*kinB* single and Δ*rhlR* Δ*kinB* double mutants failed to form colony biofilms and instead exhibited smooth colony phenotypes (Fig 2A). Introduction of a plasmid carrying the *kinB* gene conferred a hyper-rugose phenotype to the WT and restored colony biofilm formation to the Δ*kinB* and Δ*rhlR* Δ*kinB* mutants (Fig 2A). By contrast, introduction of a plasmid carrying *rhlR* did not alter the smooth biofilm phenotype of the Δ*rhlR* Δ*kinB* double mutant (Fig 2A). We conclude that in *P. aeruginosa*, KinB is essential for biofilm formation, KinB is an activator of biofilm formation, and KinB functions downstream of RhlR in the biofilm formation process.

PA14 requires Pel, the primary biofilm matrix exopolysaccharide for biofilm formation [29] (Note: PA14 does not produce the Psl exopolysaccharide, and alginate does not contribute significantly to the PA14 biofilm matrix, unlike in *P. aeruginosa* PAO1 [30]). To examine whether the mechanism by which KinB alters biofilm formation is by changing Pel production, we performed quantitative RT-PCR analyses on WT and Δ*kinB* colony biofilms, probing for the expression of the housekeeping gene *rpoD* as a control and the Pel biosynthetic gene *pelA* (Fig 2B). Expression of *rpoD* did not change between the WT and the Δ*kinB* mutant, while transcription of *pelA* was approximately 14-fold lower in the Δ*kinB* strain than in the WT. We conclude that KinB activates Pel production, which is why KinB is required for PA14 biofilm formation.

KinB is a transmembrane HK that undergoes autophosphorylation and then transfers the phosphate to its partner RR AlgB [23]. To determine whether AlgB functions downstream of KinB to control colony biofilm formation, we introduced a stop codon in place of the codon specifying residue 10 in the *algB* gene to obtain an *algB*$^{STOP}$ mutant. We note that the *algB*$^{STOP}$ mutation does not alter *kinB* expression (S1B Fig). The *algB*$^{STOP}$ mutant had a colony biofilm phenotype indistinguishable from the WT (Fig 2A). However, introduction of the *algB*$^{STOP}$ mutation into the Δ*kinB* strain restored colony biofilm formation (Fig 2A). Furthermore, overexpression of *algB* repressed colony biofilm formation in the WT, as evidenced by the resulting smooth colony biofilm phenotype (Fig 2A). Overexpression of *algB* also repressed colony

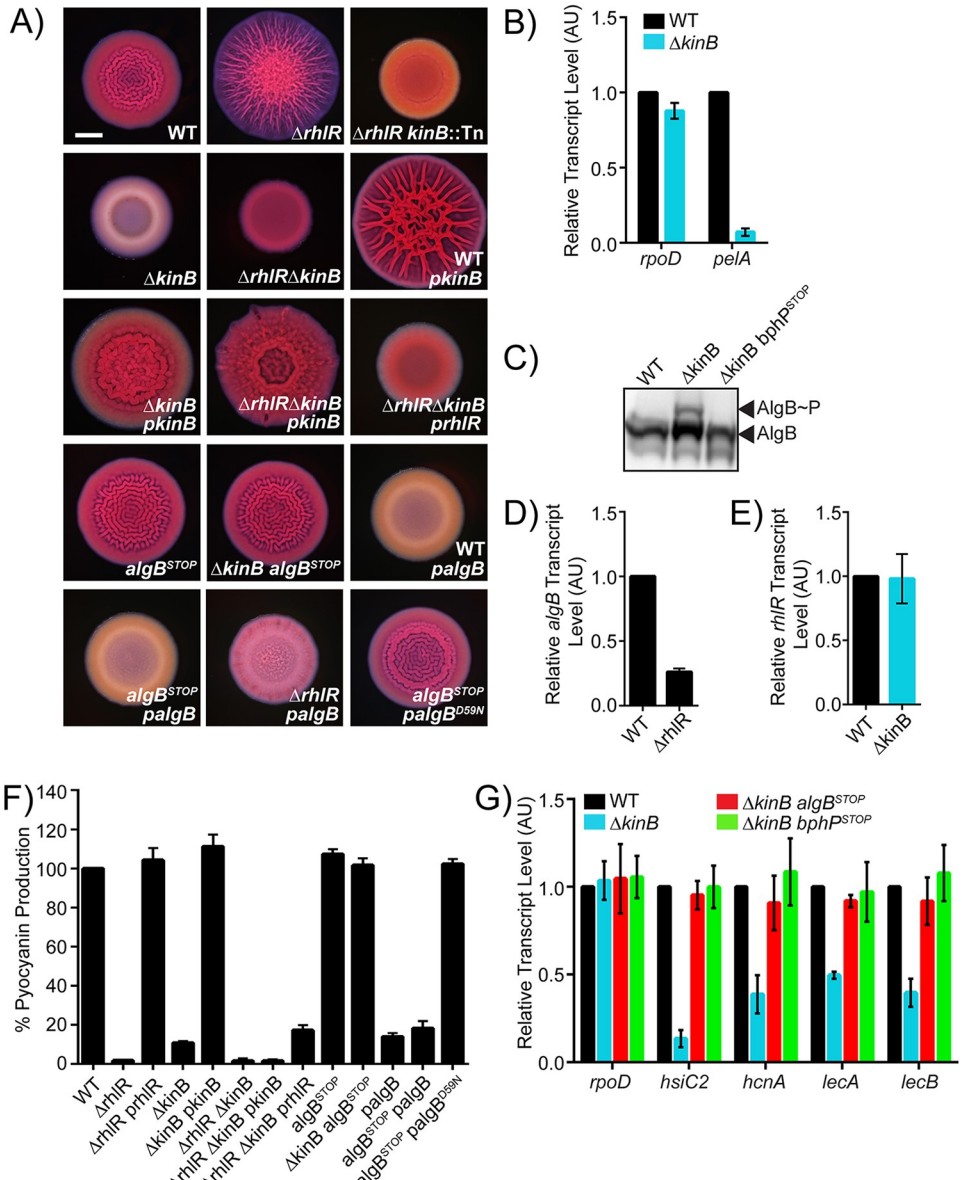

**Fig 2. RhlR represses biofilm formation via KinB.** (A) Colony biofilm phenotypes of WT PA14 and the designated mutants on Congo red agar medium after 72 h of growth. *kinB*::Tn refers to a mutant identified in a genetic screen harboring a transposon insertion in *kinB*. p*kinB*, p*rhlR*, and p*algB* refer to *kinB*, *rhlR*, and *algB*, respectively, under the $P_{lac}$ promoter on pUCP18. Scale bar for all images is 2 mm. (B) Relative expression levels of *rpoD* and *pelA* measured by qRT-PCR in WT and Δ*kinB* mutant colony biofilms grown as in (A). The *pelABCDEFG* operon encodes proteins required for the production of the Pel polysaccharide, which is essential for the formation of pellicles, SSA biofilms, and colony biofilms [29]. (C) Phos-tag western blot analysis of the indicated strains probed for 3xFLAG-AlgB. (D) Relative *algB* transcript levels measured by qRT-PCR in WT PA14 and the Δ*rhlR* mutant grown planktonically to $OD_{600} = 1.0$. (E) Relative *rhlR* transcript levels measured by qRT-PCR in WT PA14 and the Δ*kinB* mutant grown planktonically to $OD_{600} = 1.0$. (F) Pyocyanin production ($OD_{695}$) was measured in WT PA14 and the designated mutants. Production from the WT was set to 100%. (G) Relative expression of *rpoD*, *hsiC2*, *hcnA*, *lecA*, and *lecB* measured by qRT-PCR in WT PA14 and the designated mutants grown planktonically to $OD_{600} = 1.0$. *rpoD* is used as the control for comparison. For panels B, D, E, and G, data were normalized to 16S RNA levels, and the WT levels were set to 1.0. For data in panels B, D, E, F, and G, error bars represent SEM for 3 biological replicates. Data for panels B, D, E, F, and G can be found in supplemental file S1 Data. The original western blot showing the data for panel C is available in supplemental file S2 Data. AU, arbitrary unit; OD, optical density; qRT-PCR, quantitative Reverse Transcriptase-Polymerase Chain Reaction; SEM, standard error of the mean; SSA, solid-surface–associated; WT, wild type.

biofilm formation in the *algB^STOP* and Δ*rhlR* strains (Fig 2A). Thus, these data suggest that KinB activates while AlgB represses biofilm development.

AlgB has an amino-terminal domain containing the site of phosphorylation (residue D59), a central ATP-binding domain, and a carboxy-terminal helix-turn-helix motif for binding DNA (S1C and S2 Figs) [23,31]. AlgB is a member of the NtrC subfamily of RRs, and it possesses the hallmark GAFTGA motif required for interaction with RpoN (σ^54) [32]. Typically, NtrC-type RRs act as transcriptional activators when they are phosphorylated [33]. To investigate whether phosphorylation of AlgB is required for the repression of colony biofilm formation, we substituted the aspartate at residue 59 with an asparagine residue to preclude phosphorylation [32]. We overexpressed the *algB^D59N* allele in the PA14 strain carrying the *algB^STOP* mutation. Unlike WT AlgB, AlgB^D59N failed to repress colony biofilm formation (Fig 2A, S3A, S3B and S4A Figs). To ensure the validity of this result, we generated amino-terminal 3xFLAG-tagged *algB* and *algB^D59N* fusions and expressed them from a plasmid in the *algB^STOP* mutant. A western blot showed that both proteins are produced at similar levels (S3A Fig). We also introduced the *algB^D59N* mutation into the *algB* gene in the chromosome of WT PA14 and the Δ*kinB* mutant. Identical to the results from the plasmid-based experiment, both the *algB^D59N* single and the Δ*kinB algB^D59N* double mutants produced colony biofilms (S3B Fig). We conclude that the phosphorylated form of AlgB is active and is required for AlgB-mediated repression of biofilm development. We presume that AlgB-P functions indirectly as a transcriptional activator to promote the expression of a gene encoding a negative regulator of biofilm formation (Fig 1).

Our results show that AlgB functions downstream of KinB and that KinB and AlgB have opposing activities with respect to PA14 biofilm formation. In vitro, KinB possesses both kinase and phosphatase activities [25]. One mechanism by which KinB could antagonize AlgB function is by acting as a phosphatase that dephosphorylates AlgB, rendering it inactive. To test this possibility, we integrated the 3xFLAG-tagged *algB* allele at the native *algB* locus in the chromosomes of WT PA14 and the Δ*kinB* mutant. Colony biofilm analyses show that 3xFLAG-AlgB is functional (S3B Fig). Next, we assessed the phosphorylation status of 3xFLAG-AlgB in vivo. Fig 2C shows that AlgB-P accumulates in the Δ*kinB* mutant compared to in the WT. To verify these claims regarding the signal transduction mechanism, we engineered a missense mutation into KinB at a conserved proline (P390) that is required for phosphatase activity [25]. Specifically, we generated both *kinB-SNAP* and *kinB^P390S-SNAP* fusions and introduced these alleles at the native *kinB* locus on the chromosome of PA14. Carboxy-terminal tagging of KinB with SNAP does not interfere with its function because the strain carrying *kinB-SNAP* forms biofilms that are indistinguishable from those of WT PA14 (S3C and S3D Fig). The KinB^P390S-SNAP protein is also produced and stable (S3C Fig); however, identical to the Δ*kinB* mutant, the strain carrying *kinB^P390S-SNAP* fails to form colony biofilms (S3D Fig). These data demonstrate that KinB acts as a phosphatase to inhibit AlgB function in vivo. We therefore hypothesize—and we come back to this point below—that some other HK must phosphorylate AlgB to activate it and enable it to function as a repressor of biofilm development.

Our data show that the KinB–AlgB TCS functions downstream of RhlR to repress biofilm formation. An obvious mechanism by which RhlR could influence KinB–AlgB activity is by activating transcription of the *algB–kinB* operon. Indeed, RT-PCR shows that *algB* transcript levels are approximately 4-fold higher in the WT than in the Δ*rhlR* mutant (Fig 2D). Thus, RhlR activates expression of the *algB–kinB* operon. By contrast, deletion of *kinB* has no effect on *rhlR* transcript levels (Fig 2E), confirming their epistatic relationship.

KinB has been reported to be required for pyocyanin production [25]. Pyocyanin is a RhlR-dependent virulence factor [21,34]. Our findings of a regulatory connection between

KinB and RhlR suggest that KinB and RhlR could jointly regulate pyocyanin production. To test this idea, we measured pyocyanin production in planktonic cultures of WT, Δ*rhlR*, Δ*kinB*, and Δ*rhlR* Δ*kinB* strains. Similar to what has been reported previously, deletion of *rhlR* and/or *kinB* abolished pyocyanin production (Fig 2F). Overexpression of *rhlR* in the Δ*rhlR* strain and overproduction of *kinB* in the Δ*kinB* strain restored pyocyanin production, demonstrating that our expression constructs are functional (Fig 2F). By contrast, overexpression of either *rhlR* or *kinB* in the Δ*rhlR* Δ*kinB* double mutant failed to rescue pyocyanin production (Fig 2F). Thus, RhlR and KinB are both required activators of pyocyanin production in PA14. Consistent with AlgB functioning as the RR for KinB, inactivation of AlgB (i.e., *algB*$^{STOP}$) in the Δ*kinB* background restored WT levels of pyocyanin production, while overexpression of *algB* in the WT and the *algB*$^{STOP}$ mutant reduced pyocyanin levels (Fig 2F). Lastly, unlike overexpression of *algB*, overexpression of *algB*$^{D59N}$ failed to repress pyocyanin production, suggesting that phosphorylation of AlgB is required for AlgB activity (Fig 2F).

To further explore the role of the KinB–AlgB TCS on RhlR-driven gene expression, we quantified expression of 4 other RhlR-activated genes [21]—*hsiC2* (type-VI secretion), *hcnA* (hydrogen cyanide synthase), *lecA* (galactose-binding lectin), and *lecB* (fucose-binding lectin), all encoding virulence factors—in the WT and the Δ*kinB* mutant. Expression of all 4 genes was lower in the Δ*kinB* mutant than in the WT (Fig 2G). Introduction of the *algB*$^{STOP}$ mutation into the Δ*kinB* mutant restored expression of all 4 virulence genes to WT levels (Fig 2G). Thus, both RhlR and KinB activate virulence gene expression in *P. aeruginosa*. Moreover, we conclude that AlgB is epistatic to KinB for all the phenotypes tested here, and thus, KinB and AlgB function in the same pathway, albeit in opposing manners, to control biofilm formation and virulence factor production.

## The bacteriophytochrome BphP is the HK required to activate AlgB to mediate repression of quorum-sensing–controlled behaviors

We have invoked the existence of a putative HK to activate AlgB via phosphorylation. To identify this component, we used genetic suppressor analysis, reasoning that mutants with defects in the upstream component required to phosphorylate AlgB would render AlgB nonfunctional. We further reasoned that such suppressor mutants would transform the Δ*kinB* smooth colony biofilm phenotype back to the rugose phenotype because in such mutants, AlgB could not act as a repressor of biofilm formation. We isolated 12 spontaneously arising rugose mutants from Δ*kinB* smooth colony biofilms and analyzed them by whole-genome sequencing (Fig 3A). Eight suppressors contained deletions or missense mutations in the *algB* gene, while the remaining 4 suppressors harbored mutations in the *bphP* gene (Fig 3B, S2 Table). *bphP* is located in a dicistron immediately downstream of *bphO* (Fig 3B, S1 Fig) [35]. We discuss *bphO* below; here we focus on *bphP*. Exactly analogous to mutation of *algB*, mutation of *bphP* was epistatic to *kinB* for all of the phenotypes tested. Specifically, engineering a stop codon in place of the codon specifying residue 50 in the *bphP* gene showed no effect in WT PA14, but it restored colony biofilm formation, pyocyanin production, and virulence gene expression to the Δ*kinB* mutant (Figs 3C–3D and 2G). Consistent with BphP being required to activate AlgB, unlike in the WT, in the *bphP*$^{STOP}$ mutant, overexpression of *algB* failed to repress colony biofilm formation and pyocyanin production (Fig 3C and 3D). Furthermore, while overexpression of *bphP* in the WT reduced pyocyanin production to the levels of the Δ*kinB* mutant, overexpression of *bphP* had no effect in the *algB*$^{STOP}$ mutant (Fig 3D). There is a severe growth defect associated with the overexpression of *bphP*. For this reason, in Fig 3D, rather than using plasmid pUCP18, we expressed *bphP* from the low-copy–number plasmid pBBR1-MCS5. Unfortunately, the presence of the empty pBBR-MCS5 plasmid in WT and

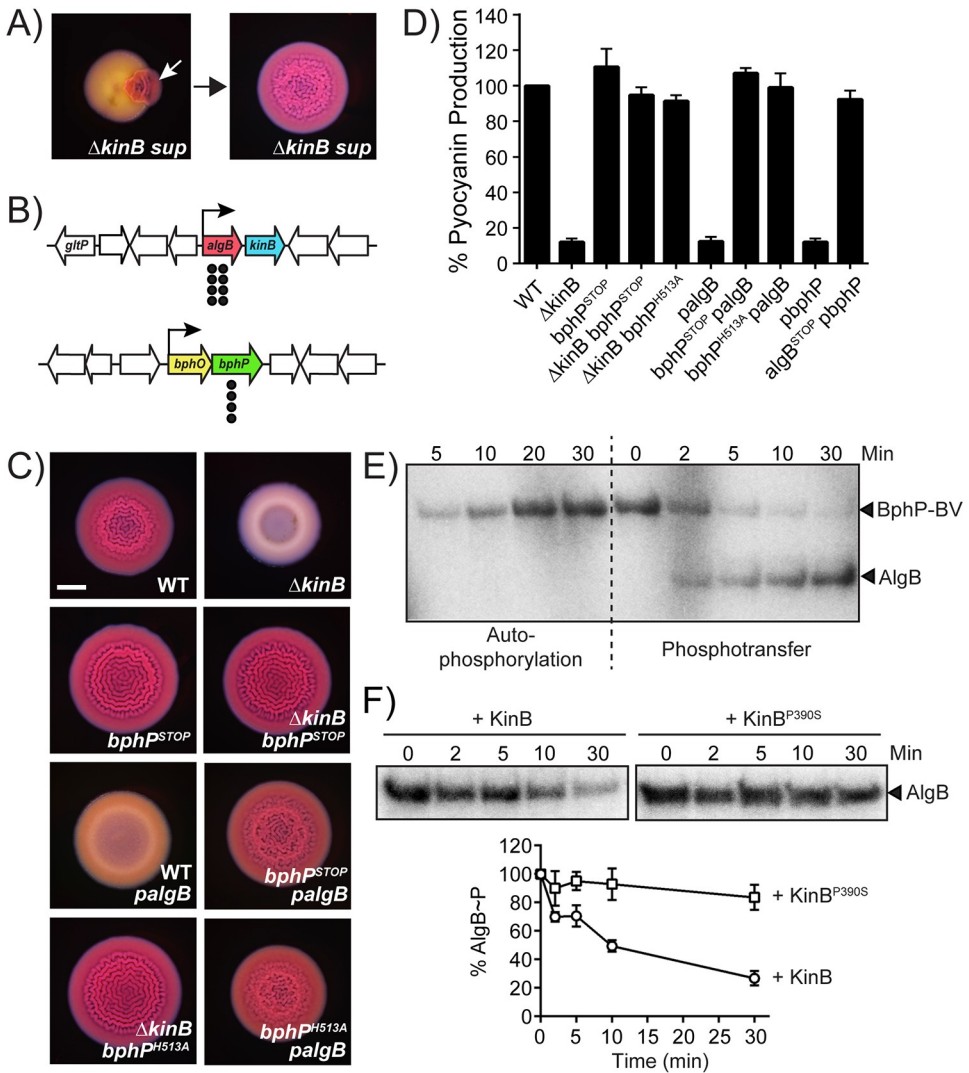

**Fig 3. BphP and AlgB are a two-component HK–RR pair.** (A) Shown is a representative isolation of a suppressor mutation of the Δ*kinB* smooth colony biofilm phenotype. The white arrow in the left panel indicates a region of rugose sectoring in the Δ*kinB* smooth colony biofilm that is diagnostic of the emergence of a suppressor mutation. The right panel shows the colony biofilm phenotype of a mutant following isolation. (B) Chromosomal arrangements of the *algB* (red), *kinB* (blue), *bphO* (yellow), and *bphP* (green) genes. Large white arrows represent open reading frames (lengths not to scale), black bent arrows indicate promoters, and black circles indicate the locations of suppressor mutations. (C) Colony biofilm phenotypes of WT PA14 and the designated mutants on Congo red agar medium after 72 h of growth. *palgB* refers to *algB* under the $P_{lac}$ promoter on the pUCP18 plasmid. Scale bar is 2 mm for all images. (D) Pyocyanin production ($OD_{695}$) was measured in WT PA14 and the designated mutants. *pbphP* refers to *bphP* under the $P_{lac}$ promoter on the pBBR-MCS5 plasmid. Error bars represent SEM for 3 biological replicates. (E) Autophosphorylation of BphP–BV and phosphotransfer to AlgB. (Left) Autophosphorylation of BphP–BV was carried out for 30 min and samples were removed at the indicated times for electrophoresis. (Right) An equimolar amount of AlgB was added to phospho-BphP-BV for 30 min and samples were removed at the indicated times for electrophoresis. (F) Dephosphorylation of AlgB-P by KinB or $KinB^{P390S}$. Phosphotransfer to AlgB from phospho-BphP-BV was carried out for 30 min. ATP was removed from the reaction, and either KinB or $KinB^{P390S}$ was added. Samples were removed at the indicated times for electrophoresis. The top panel shows representative images of gels. The bottom graph shows percent AlgB-P levels at each time point with SEM for 3 independent replicates. Band intensities for AlgB-P when KinB was added (circles) and when $KinB^{P390S}$ was added (squares) were normalized to the levels at time 0. To assess the quality of protein preparations used in panels E and F, see S4B Fig. Data for the graphs in panels D and F can be found in supplemental file S1 Data. The original autoradiographs with the data for panels E and F are available in supplemental file S2 Data. BV, biliverdin; OD, optical density; SEM, standard error of the mean; WT, wild type.

mutant PA14 strains abrogates colony biofilm formation, so we could not perform the companion colony biofilm assay to test overexpression of *bphP*. Nonetheless, we can conclude from Fig 3C and 3D that BphP is necessary and sufficient to activate AlgB.

BphP is a bacteriophytochrome that assembles with its chromophore BV, which is produced by the heme oxygenase BphO (Fig 3B, S1A and S1D Fig) to generate a photosensing HK that is activated by light [35]. *P. aeruginosa* BphP contains the HDLRNPL motif, which often contains the histidine residue that undergoes autophosphorylation in transmembrane HKs [36]. In *P. aeruginosa* BphP, this histidine is residue 513 (S1D Fig). To determine whether BphP kinase activity is required for AlgB activation, we generated the *bphP*$^{H513A}$ mutation, fused it to 3x*FLAG*, and introduced it onto the chromosome of the Δ*kinB* mutant. The BphP$^{H513A}$-3xFLAG protein is produced and stable (S3E Fig), and identically to the *bphP*$^{STOP}$ allele, the *bphP*$^{H513A}$ mutation restored colony biofilm formation and pyocyanin production to the Δ*kinB* mutant (Fig 3C and 3D). Moreover, overexpression of *algB* in the *bphP*$^{H513A}$ mutant failed to repress colony biofilm formation and pyocyanin production (Fig 3C and 3D). These results show that BphP H513 and AlgB D59 are required for signal transmission, and the signal is presumably phosphorylation.

To assess phosphorelay between BphP and AlgB, we used our 3xFLAG-AlgB in vivo construct. In addition to introducing it into the chromosome of WT PA14, we engineered it onto the chromosome of the *bphP*$^{STOP}$ mutant. Consistent with BphP being the kinase for AlgB, Fig 2C shows that the Δ*kinB bphP*$^{STOP}$ mutant lacks the band corresponding to AlgB-P. These data suggest that BphP transfers phosphate to AlgB. To verify this finding, we performed in vitro phosphotransfer assays. We purified recombinant BphP and formed a complex with it and commercially available BV to obtain the BphP–BV chromoprotein. Upon incubation with radiolabeled ATP under ambient light, BphP–BV underwent autophosphorylation (Fig 3E). BphP–BV readily transferred radiolabeled phosphate to purified AlgB but not to AlgB$^{D59N}$ (Fig 3E, S4A Fig). Purified apo-BphP and purified apo-BphP$^{H513A}$ complexed with BV both failed to autophosphorylate and thus could not transfer phosphate to AlgB (S4A Fig). Together, these data show that BphP–BV phosphorylates and thereby activates AlgB.

Our data suggest that KinB dephosphorylates AlgB, while BphP phosphorylates AlgB. To directly test this hypothesis, we reconstituted the BphP–AlgB–KinB phosphorelay in vitro. We purified the recombinant KinB and KinB$^{P390S}$ proteins and added them separately, at equimolar concentration, to AlgB-P prephosphorylated by BphP–BV. Fig 3F shows that over time, KinB dephosphorylates AlgB while AlgB-P levels remain unchanged in the presence of KinB$^{P390S}$. As control experiments, we added either KinB or KinB$^{P390S}$ to AlgB in the presence of ATP but in the absence of BphP–BV. Both KinB and KinB$^{P390S}$ underwent autophosphorylation and transferred phosphate to AlgB in vitro, but only WT KinB acted as a phosphatase to dephosphorylate AlgB (S5A–S5D Fig). Although our findings show that KinB is a dual kinase/phosphatase, under our in vivo conditions, only KinB phosphatase activity was detected. Perhaps KinB can function as a kinase for AlgB when its stimulus is present (Fig 1). Identifying the natural signal that drives the KinB kinase activity is the subject of our future work. We conclude that BphP–AlgB–KinB forms a regulatory module in which the RR AlgB is activated by the kinase activity of the HK BphP and inhibited by the phosphatase activity of the HK KinB.

## BphP-mediated photosensing represses *P. aeruginosa* quorum-sensing–controlled behaviors

The *P. aeruginosa* BphP bacteriophytochrome has been studied in vitro, and its kinase activity is reported to be activated by light [35]. To explore whether BphP photosensing has any effect on AlgB-controlled group behaviors in vivo, we compared colony biofilm formation by WT,

$\Delta kinB$, $\Delta kinB$ $bphP^{STOP}$, and $\Delta kinB$ $algB^{STOP}$ PA14 strains in the dark and under different light conditions. We note that all of the colony biofilm experiments in the previous sections were performed under ambient light. First, we consider WT PA14 and the $\Delta kinB$ mutant in the no light condition. Fig 4A shows that in the dark, both strains formed colony biofilms that were indistinguishable from one another. We interpret these results to mean that in the absence of light, the BphP kinase is inactive in both WT PA14 and the $\Delta kinB$ mutant, and AlgB is not phosphorylated, so it too is inactive. Thus, no repression of colony biofilm formation occurs (Fig 1). Now, we address the results under ambient light. WT PA14 formed colony biofilms, but the $\Delta kinB$ strain did not (Fig 4A). Our interpretation is that in the WT, ambient light activates the BphP kinase, and phosphotransfer to AlgB occurs. However, the opposing KinB phosphatase activity strips the phosphate from AlgB, thereby eliminating AlgB-dependent repression of colony biofilm formation. Thus, WT PA14 forms colony biofilms under ambient light. In the case of the $\Delta kinB$ mutant, since there is no KinB phosphatase present, ambient light is sufficient to drive BphP-mediated phosphorylation of AlgB, AlgB-P accumulates, and it represses colony biofilm formation. Based on these results, we infer that the presence or absence of light can alter group behaviors such as colony biofilm formation in *P. aeruginosa.*

Ambient light is a composite of different wavelengths of light. The PA14 BphP bacteriophytochrome is reported to be a far-red-light–sensing HK in vitro [26]. We wondered whether a particular wavelength of light could maximally activate the BphP kinase activity in vivo, and if so, perhaps, under that condition, the BphP kinase activity could override the KinB phosphatase, enabling light to repress biofilm formation in WT PA14. To test this notion, we exposed PA14 strains to blue, red, and far-red light and monitored colony biofilm formation. In contrast to WT PA14, the $\Delta kinB$ mutant failed to form colony biofilms under blue, red, and far-red light, suggesting that BphP is a promiscuous photoreceptor that is activated by far-red light and also by blue and red light (Fig 4A). Indeed, BphP phosphorylates AlgB in vitro under ambient (as above), blue, red, and far-red light, but it does not do so in the absence of light (Fig 4B). Importantly, when WT PA14 was exposed to far-red light, it failed to make colony biofilms, but rather, exhibited the smooth colony phenotype, identical to the $\Delta kinB$ mutant (Fig 4A). We conclude that far-red light is the preferred wavelength for BphP and is sufficient to repress biofilm formation in WT *P. aeruginosa.* Finally, we show that light-mediated repression of biofilm formation requires functional BphP and AlgB because the $bphP^{STOP}$ and $algB^{STOP}$ single mutants and the $\Delta kinB$ $bphP^{STOP}$ and $\Delta kinB$ $algB^{STOP}$ double mutants did not repress colony biofilm formation under the conditions tested (Fig 4A, S6A Fig). To extend our results concerning light-mediated repression of biofiolm formation, we assessed the effects of light on solid-surface–associated (SSA) biofilm formation [29] by the WT, $\Delta kinB$, $\Delta kinB$ $algB^{STOP}$, and $\Delta kinB$ $bphP^{STOP}$ PA14 strains (S6B Fig). The results exactly mirror our findings for colony biofilms: far-red light repressed SSA biofilm formation in the WT. The $\Delta kinB$ mutant exhibited a severe decrease in SSA biofilm formation under ambient and far-red light, but it produced SSA biofilms in the dark. Lastly, both the $\Delta kinB$ $algB^{STOP}$ and $\Delta kinB$ $bphP^{STOP}$ mutants formed SSA biofilms under all conditions.

One mechanism by which light could suppress biofilm formation via BphP–AlgB is by decreasing the production of the Pel exopolysaccharide. To quantify the effect of light on *pel* expression and, in turn, on biofilm formation, we measured *pelA* transcript levels using quantitative RT-PCR analyses of WT, $\Delta kinB$, $\Delta kinB$ $algB^{STOP}$, and $\Delta kinB$ $bphP^{STOP}$ colony biofilms in darkness and under ambient and far-red light (Fig 4C). We used *rpoD* transcription as the control. Expression of *rpoD* did not change under any condition tested. Regarding *pelA*, analogous to what occurred for colony biofilm formation, there was no significant difference in *pelA* expression between the WT and the $\Delta kinB$ strains in the dark, whereas transcription of *pelA* was approximately 14-fold lower in the $\Delta kinB$ strain than in the WT under ambient light.

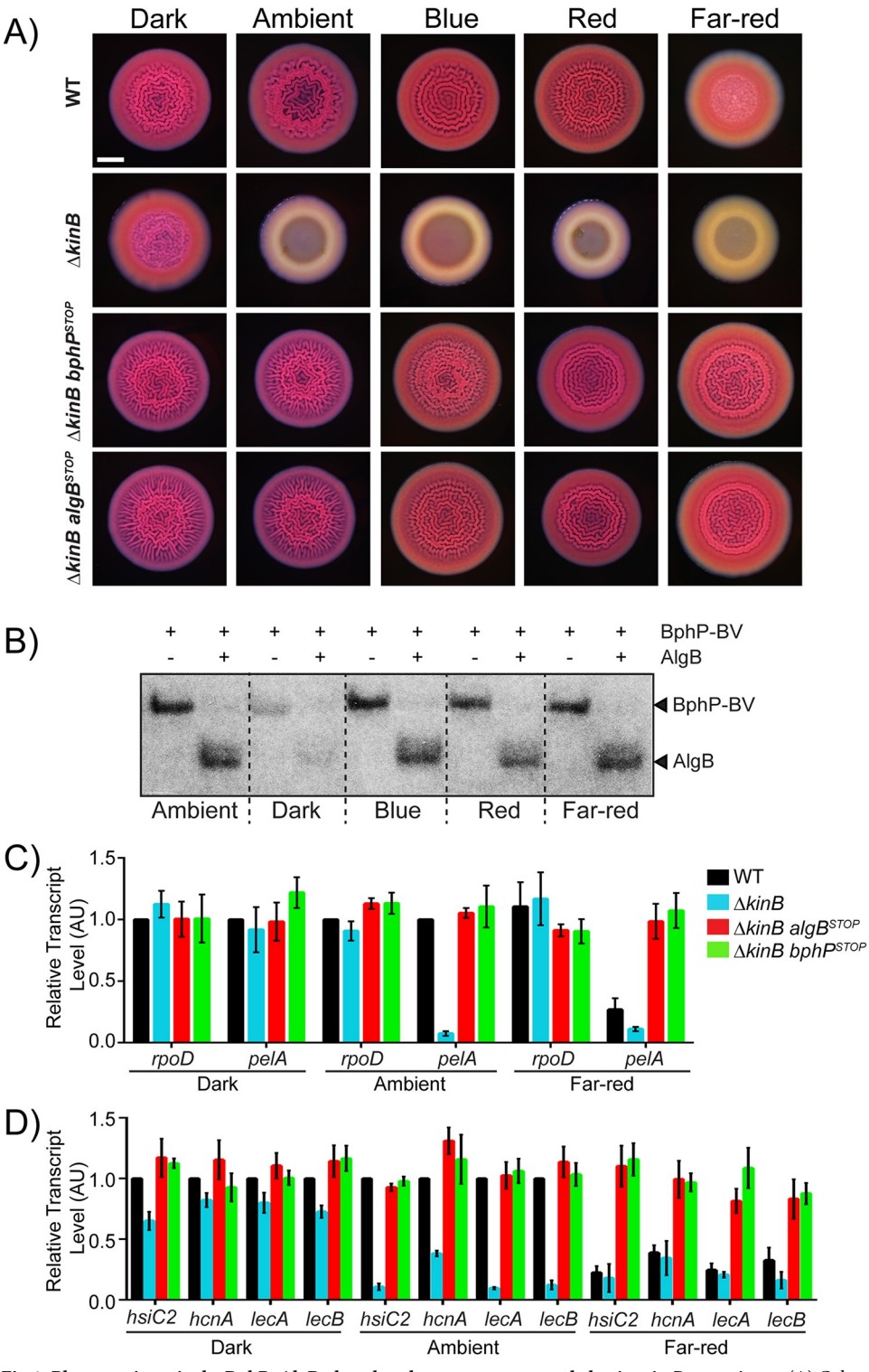

**Fig 4. Photosensing, via the BphP–AlgB phosphorelay, represses group behaviors in *P. aeruginosa*.** (A) Colony biofilm phenotypes are shown for WT PA14 and the designated mutants on Congo red agar medium after 72 h of growth under the indicated light conditions. Scale bar is 2 mm for all images. (B) Autophosphorylation of the BphP–BV complex was carried out for 30 min (left lane in each pair), followed by addition of AlgB (right lane in each pair) for an additional 30 min under the indicated light conditions. (C) Relative expression of *rpoD* and *pelA* as measured by qRT-PCR in WT PA14 and the designated mutant strains grown as colony biofilms as in (A) in darkness, ambient light, and far-red light. (D) Relative expression of *hsiC2*, *hcnA*, *lecA*, and *lecB* measured by qRT-PCR in WT PA14 and

the designated mutants grown as colony biofilms as in (A) and light conditions as in (B). For panels B and C, data were normalized to 16S RNA levels, and the WT levels were set to 1.0. Error bars represent SEM for 3 biological replicates. Data for panels C and D can be found in supplemental file S1 Data. The original autoradiograph with the data for panel B is available in supplemental file S2 Data. AU, arbitrary unit; BV, biliverdin; qRT-PCR, quantitative Reverse Transcriptase-Polymerase Chain Reaction; SEM, standard error of the mean; WT, wild type.

Repression of *pelA* expression depended on functional BphP and AlgB because the Δ*kinB bphP*$^{STOP}$ and Δ*kinB algB*$^{STOP}$ mutants transcribed *pelA* at high levels under both conditions. We conclude that dephosphorylation of AlgB does not occur in the Δ*kinB* mutant under ambient light. In this condition, BphP phosphorylates AlgB, and AlgB-P represses colony biofilm formation via down-regulation of *pelA* expression. Lastly, in the WT, *pelA* transcript levels were approximately 4-fold lower under far-red light than in darkness. Therefore, far-red light is the strongest activator of BphP such that under far-red light, but not ambient light, the kinase activity of BphP overrides the phosphatase activity of KinB in the WT to drive AlgB-P accumulation, repression of *pelA* expression, and consequently, repression of biofilm formation.

In Fig 2, we showed that BphP is required for AlgB-dependent repression of virulence gene expression. Our results in Fig 4 suggest that light, by controlling BphP-dependent phosphorylation of AlgB, could control virulence in *P. aeruginosa*. To explore this idea further, we quantified the expression of the virulence-associated genes *hsiC2*, *hcnA*, *lecA*, and *lecB* in colony biofilms of WT PA14 and in the Δ*kinB*, Δ*kinB algB*$^{STOP}$, and Δ*kinB bphP*$^{STOP}$ strains under darkness, ambient light, and far-red light. The results mimic those for biofilm formation and *pelA* transcription. Only in the absence of the opposing KinB phosphatase activity is ambient light sufficient to activate BphP, whereas far-red-light–driven BphP kinase activity can override the KinB phosphatase activity, allowing accumulation of AlgB-P to levels that repress virulence gene expression. Again, light-mediated repression of virulence genes requires functional BphP and AlgB (Fig 4D). We conclude that BphP-dependent photosensing represses virulence gene expression in *P. aeruginosa*.

Light possesses both color (wavelength) and intensity properties. Above, we demonstrated that BphP can detect blue, red, and far-red light. To explore the possibility that *P. aeruginosa* BphP is also capable of detecting light intensity, we varied the intensity of far-red light since it has the most dramatic effect on PA14 phenotypes. We used repression of colony biofilm formation as the readout. Colony biofilm formation decreased with increasing intensity of far-red light in the WT and Δ*kinB* mutant but remained unaltered in the *bphP*$^{STOP}$ mutant (Fig 5A). The highest intensity of light we tested (bottom-most panel in Fig 5A) is similar to that present in natural sunlight (5.5 W/m$^2$ in a 5-nm window around 730 nm; ASTM G173-03 Reference Solar Spectra, www.astm.org). At this intensity, WT colony biofilm formation was maximally repressed, showing that BphP kinase dominates over KinB phosphatase. The Δ*kinB* mutant generates suppressor flares under this condition, suggesting that one role of the KinB phosphatase is to keep the BphP kinase activity in check. To verify that far-red light specifically altered colony biofilm behavior without affecting general physiology, we quantified *rpoD* and *pelA* transcript levels in the WT and *bphP*$^{STOP}$ mutant colony biofilm samples grown under the different light intensities (Fig 5B). Expression of *rpoD* did not change under any condition tested, while transcription of *pelA* decreased progressively in the WT with increasing intensity of far-red light. At the highest intensity of far-red light tested, expression of *pelA* was approximately 12-fold lower than that in the *bphP*$^{STOP}$ mutant that cannot convey the light cue internally to AlgB. These results demonstrate that *P. aeruginosa* colony biofilm formation can be modulated simply by tuning the intensity of far-red light in which the strain is grown.

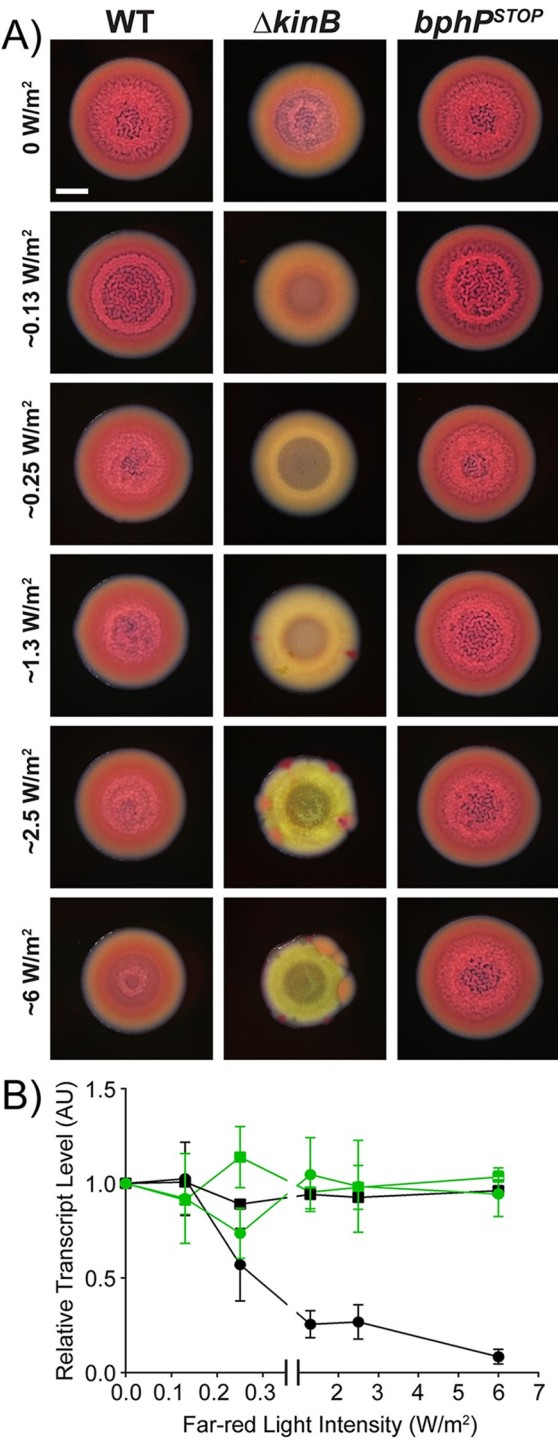

**Fig 5. Far-red light intensity controls biofilm formation.** (A) Colony biofilm phenotypes are shown for WT PA14 and the designated mutants on Congo red agar medium after 72 h of growth under the indicated far-red light intensities. Scale bar is 2 mm for all images. (B) Relative expression of *rpoD* (squares) and *pelA* (circles) measured by qRT-PCR in WT PA14 (black) and in the *bphP^STOP* mutant (green) grown as colony biofilms as in (A). Data were normalized to 16S RNA levels, and the WT levels at 0 mW/m² far-red light were set to 1.0. Error bars represent SEM for 3 biological replicates. Data for panel B can be found in supplemental file S1 Data. AU, arbitrary unit; qRT-PCR, quantitative Reverse Transcriptase-Polymerase Chain Reaction; SEM, standard error of the mean; WT, wild type.

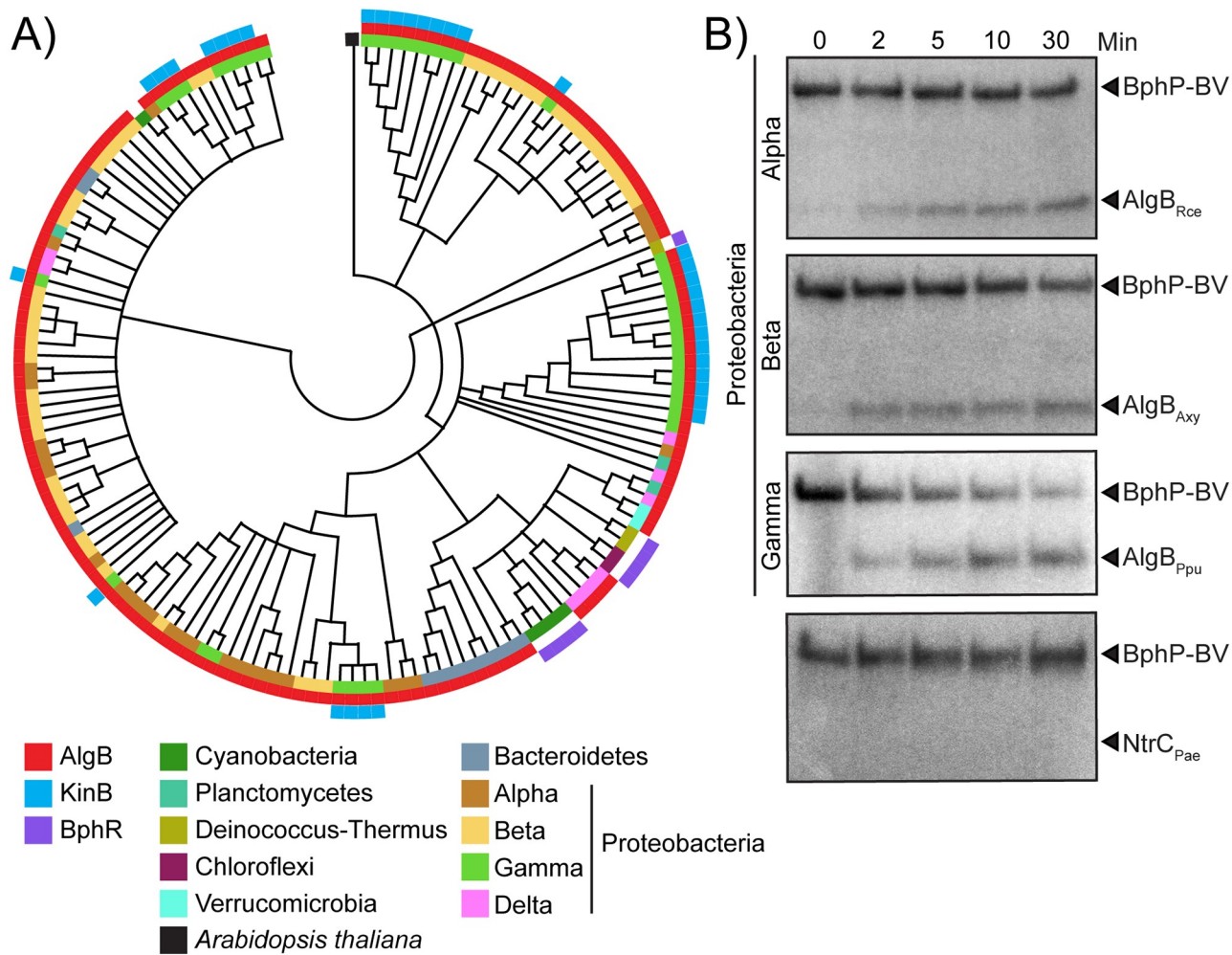

**Fig 6. The BphP–AlgB phosphotransfer relay is conserved in diverse bacteria.** (A) Maximum likelihood based phylogenetic tree for BphP showing the 150 closest orthologs to *P. aeruginosa* BphP. Co-occurrences of AlgB and KinB are depicted in red and blue, respectively. BphR is shown in purple. The other colors indicate bacterial phyla. The black square indicates *Arabidopsis thaliana* as the root of the tree. (B) In vitro phosphorylation of AlgB orthologs from the α-Proteobacterium Rce, the β-Proteobacterium Axy, and the γ-Proteobacterium Ppu by *P. aeruginosa* BphP–BV that had been autophosphorylated for 30 min. The bottom panel shows that phosphotransfer from *P. aeruginosa* phospho-BphP-BV to *P. aeruginosa* NtrC does not occur. To assess the quality of the protein preparations used in panel B, see S4B Fig. The original autoradiographs with the data for panel B are available in supplemental file S2 Data. Axy, *Achromobacter xylosocidans*; BV, biliverdin; Ppu, *P. putida*; Rce, *Rhodospirillum centenum*.

## The BphP–AlgB phosphorelay is conserved in diverse bacteria

BphP bacteriophytochromes are a major class of photoreceptors widely distributed in nonphotosynthetic bacteria [16]. These BphP HKs either lack a partner RR or, when they are co-transcribed with a partner RR gene, the physiological output of the circuit is unknown. Thus, the downstream signaling components and consequences of photosensing in nonphotosynthetic bacteria are not understood. Our discovery of AlgB as the partner RR for the orphan light-sensing BphP HK in a nonphotosynthetic bacterium, coupled with our demonstration of the biofilm and virulence outputs of photosensing, puts us in a position to test the generality of our findings. As a first step, we generated a phylogenetic tree containing 150 BphP orthologs that are the closest homologs to *P. aeruginosa* BphP (Fig 6A, S7 Fig). The majority of these BphP orthologs are present in nonphotosynthetic bacteria from diverse phyla. The pseudomonads fall into discrete clusters, hinting at acquisition of BphP via horizontal gene transfer.

With respect to AlgB and KinB, we find that while KinB is present only in the pseudomonads, *Acinetobacter baumannii*, and *Enterobacter cloacae* (Fig 6A, S1 and S7 Figs), AlgB is present in approximately 93% of the bacterial species in our BphP-based phylogenetic tree (Fig 6A, S7 Fig). We note that in all of the bacteria that do not encode AlgB, for example, *Deinococcus* spp., BphR is the partner RR for BphP (Fig 6A, S1, S2 and S7 Figs; [35]). None of these *bphP*-encoding bacteria possess both BphR and AlgB. Therefore, the pattern that emerges is that BphB is widely distributed in nonphotosynthetic bacteria and the partner RR is either AlgB or BphR.

To test whether BphP can interact with and phosphorylate AlgB in bacteria other than *P. aeruginosa*, we purified AlgB orthologs from diverse Proteobacteria: *Rhodospirillum centenum* (α), *Achromobacter xylosoxidans* (β), and *P. putida* (γ) (S2 Fig). We incubated these AlgB proteins with an equimolar concentration of autophosphorylated *P. aeruginosa* BphP–BV. Phosphotransfer from BphP–BV to the AlgB orthologs occurred in all cases, albeit to varying degrees (Fig 6B). To eliminate the possibility that phospho-BphP-BV is a promiscuous kinase for NtrC family RRs, we purified NtrC from PA14 and incubated it with autophosphorylated BphP–BV. Phospho-BphP-BV failed to phosphorylate NtrC (Fig 6B). We conclude that BphP is the specific HK for AlgB, and AlgB appears to have a conserved function in photosensory signal transduction in diverse bacteria.

## Discussion

Our study reveals that the nonphotosynthetic pathogenic bacterium *P. aeruginosa* detects and responds to light to repress group behaviors, including virulence factor production and biofilm formation. The photoreceptor BphP functions as a light-activated HK that phosphorylates the AlgB RR. AlgB-P represses group behaviors but is antagonized by its partner HK KinB. Specifically, KinB dephosphorylates AlgB, and thus, KinB functions as an activator of group behaviors. Our work shows that AlgB functions as a hub protein that has 3 inputs—quorum sensing via RhlR, photosensing via BphP, and an unknown signal via KinB. We note that *bphP* expression has been reported to be controlled by LasR-LasI quorum sensing and the stationary-phase alternative sigma factor RpoS [37], suggesting that other sensory cues can be integrated into the AlgB–KinB–BphP circuit. While quorum sensing activates *algB* expression, photosensing activates AlgB function, and thus, the presence or absence of light can override the quorum-sensing input from RhlR. We reason that at high cell density, RhlR will drive AlgB production. However, if there is no light, BphP will not phosphorylate and activate AlgB. In turn, AlgB will not repress group behaviors. To our knowledge, the BphP–AlgB photosensory signal transduction cascade represents the first example of light-mediated control of group behaviors in the global pathogen *P. aeruginosa*.

Light is a ubiquitous source of energy that drives the anabolic process of photosynthesis in photosynthetic organisms. However, the wide distribution of photoreceptors in all domains of life suggests roles for photosensing in behaviors far beyond photosynthesis. Plants, for example, use light cues to regulate activities such as seed germination [38], stomatal opening [39], and defenses against microbes [40–42]. Furthermore, plant vascular systems can function as bundles of optical fibers to efficiently transmit light, particularly far-red light, that is not absorbed by plant pigments, allowing opportunities for photosensing in roots and possibly in the rhizosphere [43]. Many of the *bphP*-encoding bacteria from the phylogenetic tree in Fig 6A that also possess AlgB are members of the rhizosphere microbiome [44]. Perhaps these nonphotosynthetic bacteria exploit light cues to colonize and/or to fine-tune their mutualistic or pathogenic interactions with their plant hosts as well as adjust their physiology in the rhizosphere. While we do not know the evolutionary forces that drove *P. aeruginosa* to become a

photosensing bacterium, we speculate below on possible advantages *P. aeruginosa* could accrue by detecting light in the environment and in the human host.

Light provides spatial and temporal information to higher organisms. Does light serve a similar purpose in bacteria? Recent studies have reported that BphP plays a role in multiple stages of infection by the foliar plant pathogens *Xanthomonas campestris pv. campestris* and *P. syringae pv. syringae* [45,46], in each case via an unknown but putative downstream RR. Based on our phylogenetic analysis, we speculate that AlgB fulfils this role. We further speculate that *P. aeruginosa*, which is a plant pathogen [47], responds to light cues via the BphP–AlgB TCS to appropriately modulate its biofilm and virulence programs, particularly to inhibit virulence during daylight, enabling avoidance of plant defense mechanisms. For instance, during the day, chlorophyll in leaves removes most of the red wavelength from sunlight but little of the far-red spectrum [48]. Thus, far-red light is readily available, and based on our work here, could signal to *P. aeruginosa* to tamp down virulence factor production and biofilm formation, allowing it to optimize those programs in line with host conditions because shaded leaves are more susceptible to infection than leaves exposed to direct light [49]. In the same vein, our work suggests the possibility that inside the mammalian host, the lack of light would drive *P. aeruginosa* to form biofilms and produce virulence factors because, in the absence of light, BphP–AlgB-mediated repression of biofilms and virulence would not occur.

In addition to providing spatial and temporal information, light can also reveal other key parameters to which bacteria respond. Detection of blue light via LOV- and BLUF-domain proteins modulates general stress responses in some nonphotosynthetic bacteria such as *Bacillus subtilis* and *Caulobacter crescentus* [50,51]. Light, through the LOV-HK of the mammalian pathogen *Brucella abortus*, is crucial for virulence in a macrophage infection model, although the components connecting light to the virulence response remain undefined. It is also proposed that *B. abortus* uses light as an indicator of whether it is inside or outside of its animal host [52]. The *P. aeruginosa* genome does not encode LOV or BLUF domain proteins [11]. *P. aeruginosa* possesses only one identifiable photoreceptor, BphP. Nonetheless, we showed that *P. aeruginosa* is capable of detecting blue, red, and far-red light via BphP (Fig 4A and 4B). Perhaps an advantage of BphP promiscuity is that it enables detection of higher energy and therefore phototoxic blue light, in addition to the lower-energy but highly penetrative far-red light. Such a scenario would endow *P. aeruginosa* with the plasticity to diversify its physiological outputs in response to particular wavelengths of light without the necessity of a distinct photoreceptor for each wavelength.

Bacteriophytochromes are thought to function as reversible photoswitches that convert between red-absorbing and far-red–absorbing states [17]. The *P. aeruginosa* BphP kinase has been reported to be activated by far-red light in vitro [26]. Interestingly, however, the published BV absorption spectrum possesses peaks at blue, red, and far-red light [53], hinting at promiscuity in light detection. Based on our in vitro and in vivo data, we suggest that *P. aeruginosa* BphP is active as a kinase in both the red and far-red states as well as under blue light. We are currently investigating the mechanism by which the BphP–BV holocomplex is activated by different wavelengths of light.

An advantage *P. aeruginosa* could gain by sensing light on or within a mammalian host would be the ability to tune into the host circadian rhythm and its associated responses. Circadian clocks influence various aspects of health and disease such as sleep/wake cycles and metabolism [54,55]. Disruption of circadian rhythms are associated with fitness costs [55]. In mammals, both innate and adaptive immune responses are controlled by the circadian clock such that the immune system is primed to combat pathogens during the host active phase while immune functions undergo regeneration and repair during the resting phase of the daily cycle. Parasites such as *Plasmodium* spp., which cause malaria, synchronize their replication

cycle with host circadian rhythms for optimized infection and dissemination [56]. Likewise, viruses such as herpes and influenza A have been shown to exploit the mammalian circadian clock for their own gain, i.e., to successfully avoid host immune responses, enabling maximal replication [57,58]. Perhaps *P. aeruginosa* uses light as a signal that reveals when the host immune response is at peak function, and accordingly, at that time, *P. aeruginosa* represses biofilm formation and virulence factor expression as a mechanism that enhances evasion of host defenses. If so, a human host infected with *P. aeruginosa* during the night would be colonized to higher levels compared to a host acquiring an infection during the day. Synchronizing infectivity with light/dark cues to enable optimal infection could be a common feature of non-photosynthetic photoreceptor-harboring pathogens.

*P. aeruginosa* is a priority pathogen on the Centers for Disease Control and Prevention (CDC) ESKAPE pathogen list (a set of bacteria including *Enterococcus faecium*, *Staphylococcus aureus*, *Klebsiella pneumoniae*, *A. baumannii*, *P. aeruginosa*, and *Enterobacter* spp. that are designated as multidrug-resistant pathogens requiring new antimicrobials for treatment) and a critical pathogen on the World Health Organization (WHO) priority list [59,60]. Our phylogenetic analysis suggests that the KinB–AlgB–BphP module is conserved in the genomes of *A. baumannii* and *E. cloacae*, perhaps acquired from *P. aeruginosa* via horizontal gene transfer because the AlgB primary sequence is nearly identical between the 3 species. We speculate that beyond *P. aeruginosa*, BphP–AlgB-dependent photosensing also affects the physiology and possibly the virulence of these ESKAPE pathogens. Collectively, the results from this study provide unanticipated insight into *P. aeruginosa* physiology and a surprising possibility for therapeutic intervention—shining light on a deadly and actively studied pathogen, *P. aeruginosa*, to attenuate virulence and biofilm formation. One can imagine such a strategy could be deployed in external infections such as burns, which are highly susceptible to *P. aeruginosa* and could, moreover, be subjected to precise light regimes.

## Materials and methods

### Bacterial strains and growth conditions

All strains and plasmids used in this study are listed in S3 and S4 Tables, respectively. *P. aeruginosa* PA14 and mutants were grown at 37 ˚C in lysogeny broth (LB) (10 g tryptone, 5 g yeast extract, 5 g NaCl per L), in 1% Tryptone broth (TB) (10 g tryptone per L), or on LB plates fortified with 1.5% Bacto agar. When appropriate, antimicrobials were included at the following concentrations: 400 μg/mL carbenicillin, 30 μg/mL gentamycin, and 100 μg/mL irgasan. *Escherichia coli* was grown at 37 ˚C in LB or on LB plates fortified with 1.5% Bacto agar and the following concentrations of antimicrobials as appropriate: 15 μg/mL gentamycin, 50 μg/mL kanamycin, and 100 μg/mL ampicillin. IPTG (Sigma-Aldrich, St. Louis, MO, USA) was added to the medium at the indicated concentrations when appropriate.

### Mutant strain and plasmid construction

Strains and plasmids were constructed as described previously [22]. Briefly, to construct markerless in-frame chromosomal deletions and substitutions in PA14, DNA fragments flanking the gene of interest were amplified, assembled by the Gibson method [61], and cloned into suicide vector pEXG2 [62]. The resulting plasmids were used to transform *E. coli* SM10λ*pir* and subsequently mobilized into PA14 strains via biparental mating. Exconjugants were selected on LB containing gentamycin and irgasan, followed by recovery of deletion mutants on LB medium containing 5% sucrose. Candidate mutants were confirmed by PCR and Sanger sequencing. Transposon insertions in the PA14 chromosome were generated by mating the PA14 Δ*rhlR* parent strain with *E. coli* SM10λ*pir* harboring pIT2 (IS*lacZ*/hah) [27]. Insertion

mutants were selected on LB agar containing 60 μg/mL tetracycline, and 100 μg/mL irgasan was included in the agar to counterselect against the *E. coli* donor. Transposon insertion locations were determined by arbitrary PCR and sequencing as described previously [27].

Protein production constructs were generated by amplifying the *algB*, *kinB*, and *bphP* coding regions and cloning them in pET28b or pET21b expression vectors (Addgene, Watertown, MA, USA) to obtain pET28b-His6-AlgB, pET21b-KinB-His6, and pET21b-BphP-His6, respectively. To generate the AlgB$^{D59N}$, KinB$^{P390S}$, and BphP$^{H513A}$ variants, the corresponding mutations were engineered onto the pET28b-His6-AlgB, pET21b-KinB-His6, and pET21b-BphP-His6 plasmids, respectively, via Gibson assembly. AlgB orthologs from *R. centenum* and *A. xylosoxidans* were amplified from gene fragments obtained from Integrated DNA Technologies (Coralville, IA, USA), and that from *P. putida* was amplified from the *P. putida* KT2440 genome. All of the gene orthologs were cloned into the pET28b plasmid.

## Pyocyanin assay

PA14 strains were grown overnight in LB liquid medium at 37 ˚C with shaking at 250 rotations per minute (rpm). The cells were pelleted by centrifugation at $21,130 \times g$ for 2 min, and the clarified supernatants were passed through 0.22-μm filters (Millipore, Burlington, MA, USA) into clear plastic cuvettes. The $OD_{695}$ of each sample was measured on a spectrophotometer (Beckman Coulter DV 730; Brea, CA, USA) and normalized to the culture cell density, which was determined by $OD_{600}$.

## Colony biofilm assay

The procedure for establishing colony biofilms has been described [22]. Briefly, 1 μL of overnight cultures of PA14 strains were spotted onto $60 \times 15$ mm Petri plates containing 10 mL 1% TB medium fortified with 40 mg/L Congo red and 20 mg/L Coomassie brilliant blue dyes and solidified with 1% agar. Colony biofilms were grown at 25 ˚C for 72 h in an incubator (Benchmark Scientific, Sayreville, NJ, USA), and images were acquired using a Leica stereomicroscope M125 (Wetzlar, Germany) mounted with a Leica MC170 HD camera at 7.78× zoom.

For colony biofilms exposed to specific wavelengths of light, the following light-emitting diodes (LEDs) were used: blue, 430 nm (Diffused RGB LED, #159; Adafruit, New York, NY, USA); red, 630 nm (Diffused RGB LED, #159; Adafruit); and far-red, 730 nm (LST1-01G01-FRD1-00; Opulent Americas, Raleigh, NC, USA). Ambient light exposure refers to biofilms grown under laboratory light conditions. For the colony biofilms shown in Fig 4A, light intensity was normalized by photon flux, and the following intensities were used: blue (0.7 W/m$^2$), red (1 W/m$^2$), and far-red (1.1 W/m$^2$). Light intensity was calibrated using a laser power meter (Ophir, North Logan, UT, USA) in a 5-nm window at the appropriate wavelength. Colony biofilm samples were grown in custom laser-cut acrylic chambers. Each chamber housed a single LED light source and an individual Petri plate containing 4 technical replicates. Samples exposed to darkness were housed in the same chambers as the light-exposed samples, but with no current applied to the LEDs.

## SSA biofilm formation assay

The procedure for establishing SSA biofilms has been described [29]. Briefly, overnight cultures of PA14 strains were diluted to a final $OD_{600}$ of 0.01 in 1% TB. These samples were used to make standing 3-mL cultures in $18 \times 150$ mm borosilicate glass tubes that were incubated (Benchmark Scientific) at 25 ˚C for 72 h under the light conditions described above for the colony biofilm assay. The cultures were poured out of the tubes, and the tubes were washed vigorously with tap water. The remaining biofilms were stained by the addition of 5 mL 0.1% crystal violet solution

into each tube. After 30 min, tubes were washed twice with tap water and left to dry overnight. Subsequently, 5 mL 33% glacial acetic acid solution was added to each tube. The crystal violet stain was quantified at $OD_{550}$ using a spectrophotometer (Beckman Coulter DV 730).

## qRT-PCR

WT PA14 and mutant strains were harvested from planktonic cultures ($OD_{600} = 1.0$) or from colony biofilms grown for 72 h. RNA was purified using the Zymo Research kit, and the preparations were subsequently treated with DNAse (TURBO DNA-free; Thermo Fisher Scientific, Waltham, MA, USA). cDNA was synthesized using SuperScript III Reverse Transcriptase (Invitrogen, Carlsbad, CA, USA) and quantified using PerfeCTa SYBR Green FastMix Low ROX (Quanta Biosciences, Beverly, MA, USA).

## Protein purification

*His6-AlgB.* The pET28b-His6-AlgB protein production vector was transformed into *E. coli* BL21 (DE3) and the culture grown to approximately 0.8 $OD_{600}$ in 1 L of LB supplemented with 50 μg/mL kanamycin at 37 ˚C with shaking at 220 rpm. Protein production was induced by the addition of 1 mM IPTG, followed by incubation of the culture for another 3 h at 25 ˚C with shaking. The cells were pelleted by centrifugation at $16,100 \times g$ for 20 min and resuspended in AlgB-lysis buffer (50 mM $NaH_2PO_4$ [pH 8.0], 300 mM NaCl, 1 mM $MgCl_2$, 1 mM DTT, 5% glycerol, 0.1% Triton X-100, 10 mM imidazole, and protease inhibitor cocktail [Roche, Basel, Switzerland]). The preparation was frozen at −80 ˚C overnight. The frozen cell pellet was thawed on ice, and the cells were lysed by sonication (1-s pulses for 15 s). The sample was subjected to centrifugation at $32,000 \times g$ for 30 min at 4 ˚C. The resulting clarified supernatant was combined with Ni-NTA resin (Novagen) and incubated for 3 h at 4 ˚C. The bead/lysate mixture was loaded onto a 1-cm separation column (Bio-Rad, Hercules, CA, USA), the resin was allowed to pack, and then it was washed with AlgB-wash buffer (50 mM $NaH_2PO_4$ [pH 8.0], 300 mM NaCl, 1 mM $MgCl_2$, 1 mM DTT, 5% glycerol, 0.1% Triton X-100, 30 mM imidazole, and protease inhibitor cocktail [Roche]). Resin-bound His6-AlgB was eluted twice with 1 mL AlgB-wash buffer containing 250 mM imidazole. Fractions were analyzed by SDS-PAGE, and the gel was stained with Coomassie brilliant blue to assess His6-AlgB purity. Purified protein was dialyzed in AlgB-storage buffer (50 mM $NaH_2PO_4$ [pH 8.0], 300 mM NaCl, 1 mM $MgCl_2$, 1 mM DTT, 5% glycerol, and 0.1% Triton X-100) and stored at −80 ˚C.

*BphP-His6.* The pET21b-BphP-His6 protein production vector was transformed into *E. coli* BL21-CodonPlus (DE3)-RIPL (Agilent Technologies, Santa Clara, CA, USA). BphP-His6 was purified as described for His6-AlgB with the following changes in buffers: BphP-lysis buffer (50 mM $NaH_2PO_4$ [pH 8.0], 300 mM NaCl, 1% Triton X-100, 0.1% β-mercaptoethanol, 10 mM imidazole, and protease inhibitor cocktail [Roche]), BphP-wash buffer (50 mM $NaH_2PO_4$ [pH 8.0], 300 mM NaCl, 1% Triton X-100, 0.1% β-mercaptoethanol, 30 mM imidazole, and protease inhibitor cocktail [Roche]), and BphP-storage buffer (50 mM $NaH_2PO_4$ [pH 8.0], 300 mM NaCl, 1% Triton X-100, 0.1% β-mercaptoethanol, 5% glycerol).

*KinB-His6.* The pET21b-KinB-His6 protein production vector was transformed into *E. coli* BL21 (DE3). KinB-His6 protein was purified exactly as described above for BphP-His6.

## Phosphorylation assays

Autophosphorylation assays were performed with purified WT BphP and the BphP[H513A] variant or with KinB and the KinB[P390S] variant. 100 μM BphP or BphP[H513A] was incubated under ambient light with 10-fold molar excess of BV (Sigma-Aldrich) for 1 h prior to the assay to form the light-activated BphP–BV stocks. Reactions were carried out in phosphorylation

buffer (50 mM Tris [pH 8.0], 100 mM KCl, 5 mM $MgCl_2$, and 10% (v/v) glycerol) and were initiated with the addition of 100 μM ATP and 2 μCi [γ-$^{32}$P]-ATP (PerkinElmer, Waltham, MA, USA). Reactions were incubated at room temperature and terminated by the addition of SDS-PAGE loading buffer. Reaction products were separated using SDS-PAGE. Gels were dried at 80 °C on filter paper under vacuum, exposed to a phosphoscreen overnight, and subsequently analyzed using a Typhoon 9400 scanner (GE Healthcare, Chicago, IL, USA) and ImageQuant software. For phosphotransfer to AlgB, an equimolar concentration of AlgB was added to the phosphorylated BphP–BV or phosphorylated KinB proteins (all proteins assayed at 5 μM). Reactions were incubated at room temperature for the indicated times and terminated by the addition of SDS-PAGE loading buffer. For the BphP–AlgB phosphorelay shown in S4B Fig, BphP–BV was incubated under specific light wavelengths using the same devices as described above for colony biofilm assay.

Dephosphorylation of AlgB-P: 10 μM AlgB was phosphorylated for 30 min in reactions containing 10 μM BphP–BV, 100 μM ATP, and 2 μCi [γ-$^{32}$P]-ATP in phosphorylation buffer. Subsequently, the reactions containing AlgB-P were applied to gel filtration spin columns (Probe Quant G-50, GE Healthcare) to remove ATP. Dephosphorylation reactions were initiated by adding 10 μM KinB or KinB$^{P390S}$. Aliquots were taken at the indicated times and analyzed as described above.

### Phos-tag SDS-PAGE and western blotting

WT PA14 and mutant strains were harvested from planktonic cultures (OD$_{600}$ = 1.0). Cells were resuspended in 100 μl of ice-cold BugBuster reagent (Novagen) containing EDTA-free Protease Inhibitor Cocktail (Roche), followed by end-over-end rotation on a nutator at room temperature for 30 min. Cell debris was removed by centrifugation (4 °C at 10,000 rpm for 1 min). 50 μL 4× SDS-PAGE loading buffer (Thermo Fisher Scientific) containing 15% β-mercaptoethanol was combined with 50 μL of the sample supernatant. Ten μL samples were loaded onto a 12.5% SuperSepTM Phos-tag gel (Wako Pure Chemical Industries, Osaka, Japan). Samples were subjected to electrophoresis at 4 °C for 3 h. Gels were incubated for 20 min on a shaking platform in 1× transfer buffer containing 1 mM EDTA and re-equilibrated for 20 min in 1× transfer buffer lacking EDTA. Proteins were transferred to nitrocellulose membranes, blocked with 5% skim milk in TBS at room temperature for 1 h, and incubated with primary anti-FLAG antibody (Sigma-Aldrich) at 1:5,000 dilution in 5% skim milk in TBS overnight at 4 °C on a rocking platform. Membranes were washed 3 times with TBS-Tween 20 at room temperature for 10 min on a rocking platform and subsequently developed with a SuperSignal West Femto Kit (Thermo Scientific) and captured with an LAS-4000 Imager (GE Healthcare).

### Whole-genome sequencing

*P. aeruginosa* strains were harvested from planktonic cultures (OD$_{600}$ = 2.0), and DNA was purified using DNeasy Blood & Tissue kit (Qiagen, Hilden, Germany). The Nextera DNA Library Prep kit (Illumina, San Diego, CA, USA) was employed with 2 ng of genomic DNA to prepare the library. Unique barcodes were added to each sample to enable multiplexing. The libraries were examined for quality using Bioanalyzer DNA High Sensitivity chips (Agilent Technologies) and quantified using a Qubit fluorometer (Invitrogen). DNA libraries from the different strains were pooled at equal molar amounts and sequenced using an Illumina MiSeq as pair-end 2 × 100 nt reads. Only the Pass-Filter (PF) reads were used for further analysis.

Whole-genome sequencing data were processed with breseq version 0.33.2 to identify mutations relative to the reference *P. aeruginosa* UCBPP-PA14 genome (www.pseudomonas.com; [63]). All high-confidence and putative SNPs and deletion events were confirmed by a manual

examination of the read pileups with GenomeViewer IGV 2.4.8. A sample collected prior to the suppressor mutation screen was aligned against the reference genome of PA14, yielding a manually curated list of 25 differences acquired by our laboratory strain prior to the experiment (19 SNPs, 6 single-nucleotide indels). Applying these differences to PA14 using gdtools (part of the breseq package) yielded an updated reference genome against which all other samples were compared. S2 Table reports all high-confidence mutations identified in this analysis.

## Phylogenetic analysis

The maximum likelihood tree for BphP orthologs was generated using MEGA-X software as described previously [64].

## Supporting information

**S1 Fig. Gene conservation for KinB, AlgB, BphO, and BphP; *kinB* expression analysis; and domain architectures of the AlgB and BphP proteins.** (A) The genes flanking *kinB*, *algB*, *bphO*, and *bphP* are diagrammed for the indicated genomes. The relative gene positions and orientations are accurate, but gene lengths are not to scale. (B) Relative expression of *kinB* measured by qRT-PCR in WT PA14 and the $algB^{STOP}$ mutant grown planktonically to $OD_{600} = 1.0$. Data were normalized to 16S RNA levels, and the WT levels were set to 1.0. Error bars represent SEM for 3 biological replicates. (C) The domain architecture of the AlgB monomer is shown. Residue 59 is required for phosphorylation; the GAFTGA motif, indicated by the magenta shading, is required for interaction with $\sigma^{54}$; and HTH refers to the helix-turn-helix DNA binding domain. Adapted from [31]. (D) Domain organization of the BphP monomer consisting of the PAS, GAF, PHY, and HK domains is shown. BV binds to the GAF domain, and residue H513 is required for autophosphorylation. Adapted from [17]. Data for panel B can be found in supplemental file S1 Data. AU, arbitrary unit; BV, biliverdin; GAF, cGMP-specific phosphodiesterases, adenylate cyclases, and FhlA; HK, histidine kinase; PAS, Per-Arnt-Sim; PHY, phytochrome; qRT-PCR, quantitative Reverse Transcriptase-Polymerase Chain Reaction; SEM, standard error of the mean.
(TIF)

**S2 Fig. Multiple sequence alignment for AlgB orthologs.** Primary sequence alignment of NtrC (first line) and AlgB (second line) from Pae and AlgB orthologs (third through twelfth lines) from Pfl, Psy, Ppr, Pst, Pen, Ppu, Aba, Ecl, Axy, Rce, and BphR (thirteenth line) from *Deinococcus radiodurans*. Highly conserved amino acids are highlighted in black. Residue 59 is shown by the green asterisk. The GAFTGA motif required for interaction with $\sigma^{54}$ is indicated by the magenta line. Aba, *Acinetobacter baumanii*; Axy, *Achromobacter xylosoxidans*; Ecl, *E. cloacae*; Pae, *P. aeruginosa*; Pen, *P. entomophila*; Pfl, *P. fluorescens*; Ppr, *P. protegens*; Ppu, *P. putida*; Pst, *P. stutzeri*; Psy, *P. syringae*; Rce, *R. centenum*.
(TIF)

**S3 Fig. $AlgB^{D59N}$, $KinB^{P390S}$, and $BphP^{H513A}$ are produced and stable in *P. aeruginosa*.** (A) Western blot analysis of whole cell lysates from the indicated strains, all of which have the $algB^{STOP}$ allele at the native locus in the genome and carry an empty vector or *3xFLAG-algB* or $3xFLAG-algB^{D59N}$ on the pBBR1-MCS5 plasmid under the $P_{lac}$ promoter. The same cell lysates were probed for RNAP as the loading control. (B) Colony biofilm phenotypes of WT PA14 and the designated mutants. Scale bar is 2 mm. (C) SDS-PAGE analysis of whole cell lysates from the indicated strains. The gel was stained for SNAP using SNAP-Cell 647-SiR fluorescent substrate (New England Biolabs, Ipswich, MA, USA). Lysozyme was added as the loading control. (D) Colony biofilm phenotypes of the *kinB-SNAP* and $kinB^{P390S}$-*SNAP* strains. Scale bar

is 2 mm. (E) Western blot analysis of whole cell lysates from the indicated strains. The same cell lysates were probed for RNAP as the loading control. The original western blots showing the data for panels A, C, and E are available in supplemental file S2 Data. RNAP, RNA Polymerase; WT, wild type.
(TIF)

**S4 Fig. Phosphotransfer from BphP to AlgB in vitro.** (A) Autophosphorylation of the BphP–BV complex was carried out for 30 min (leftmost lane), followed by addition of AlgB (second lane) or AlgB$^{D59N}$ (third lane) for an additional 30 min. The kinase-defective BphP$^{H513A}$-BV complex was incubated with radiolabeled ATP for 30 min (fourth lane), followed by addition of AlgB (fifth lane) for an additional 30 min. The apo-BphP protein was incubated with radiolabeled ATP for 30 min (sixth lane). (B) SDS-PAGE gel stained with Coomassie brilliant blue showing the indicated purified proteins. Ten μL of a 20 μM stock of each protein was loaded. The original autoradiograph showing the data for panel A is available in the supplemental file S2 Data. BV, biliverdin.
(TIF)

**S5 Fig. KinB and KinB$^{P390S}$ can phosphorylate AlgB in vitro.** (A) Autophosphorylation of KinB was carried out for 30 min, and samples were removed at the indicated times. (B) An equimolar amount of AlgB was added to KinB that had been autophosphorylated for 30 min as in (A). Samples were taken at the indicated times. (C and D) As in A and B, respectively, but for the phosphatase-deficient protein KinB$^{P390S}$. The original autoradiographs with the data for this figure are available in supplemental file S2 Data.
(TIF)

**S6 Fig. Photosensing represses *P. aeruginosa* colony biofilm formation and SSA biofilm formation.** (A) Colony biofilm phenotypes are shown for WT PA14 and the designated mutants on Congo red agar medium after 72 h of growth under the indicated light conditions. Scale bar is 2 mm for all images. (B) SSA biofilm phenotypes assessed by crystal violet staining are shown for WT PA14 and the designated mutants after 72 h of growth under the indicated light conditions. Data can be found in supplemental file S1 Data. SSA, solid-surface–associated; WT, wild type.
(TIF)

**S7 Fig. The BphP–AlgB module is conserved in diverse bacteria.** Enlarged maximum-likelihood–based phylogenetic tree for BphP from Fig 6A showing the 150 closest orthologs to *P. aeruginosa* BphP. Co-occurrences of AlgB and KinB are depicted using red and blue dots, respectively. The presence of BphR is shown by purple dots. The colored squares indicate the corresponding bacterial phyla. The black square indicates *A. thaliana* as the root of the tree.
(TIF)

**S1 Table. Transposon insertion locations.**
(DOCX)

**S2 Table. Suppressor mutations of the Δ*kinB* smooth colony biofilm phenotype.**
(DOCX)

**S3 Table. Bacterial strains used in this study.**
(DOCX)

**S4 Table. Plasmids used in this study.**
(DOCX)

**S1 Data. Excel file containing numerical data for all main and supplemental figures.**
(XLSX)

**S2 Data. PDF file containing original autoradiographs, gels, and western blots.**
(PDF)

## Acknowledgments

We thank Wei Wang and the Genomics Core Facility at Princeton University for help with whole-genome sequencing. We thank Ned Wingreen, Anne-Florence Bitbol, Joseph E. Sanfillipo, and all members of the Bassler group for thoughtful discussions.

## Author Contributions

**Conceptualization:** Sampriti Mukherjee, Bonnie L. Bassler.

**Formal analysis:** Sampriti Mukherjee, Bonnie L. Bassler.

**Funding acquisition:** Sampriti Mukherjee, Bonnie L. Bassler.

**Investigation:** Sampriti Mukherjee, Bonnie L. Bassler.

**Methodology:** Sampriti Mukherjee, Matthew Jemielita, Mikhail Tikhonov.

**Project administration:** Bonnie L. Bassler.

**Resources:** Sampriti Mukherjee, Matthew Jemielita, Vasiliki Stergioula, Bonnie L. Bassler.

**Supervision:** Bonnie L. Bassler.

**Validation:** Sampriti Mukherjee, Bonnie L. Bassler.

**Writing – original draft:** Sampriti Mukherjee, Matthew Jemielita, Mikhail Tikhonov, Bonnie L. Bassler.

**Writing – review & editing:** Sampriti Mukherjee, Mikhail Tikhonov, Bonnie L. Bassler.

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
