## [Editor Report · Decision Letter 0]

16 Sep 2019

Dear Dr Bassler, 

Thank you for submitting your manuscript entitled "Photo sensing and quorum sensing are integrated to control Pseudomonas aeruginosa collective behaviors" for consideration as a Research Article by PLOS Biology.

Your manuscript has now been evaluated by the PLOS Biology editorial staff, as well as by an academic editor with relevant expertise, and I'm writing to let you know that we would like to send your submission out for external peer review.

Please re-submit your manuscript within two working days, i.e. by Sep 18 2019 11:59PM.

Kind regards,

Roli Roberts

Senior Editor

PLOS Biology

---

## [Decision Letter · Decision Letter 1]

28 Oct 2019

Dear Dr Bassler,

Thank you very much for submitting your manuscript "Photo sensing and quorum sensing are integrated to control Pseudomonas aeruginosa collective behaviors" for consideration as a Research Article at PLOS Biology. Your manuscript has been evaluated by the PLOS Biology editors, an Academic Editor with relevant expertise, and by four independent reviewers.

You'll see that while the reviewers' assessments of your study are broadly positive, each of them raises some concerns that will need addressing, some with additional experimental data. IMPORTANT: The Academic Editor notes that some of the papers mentioned by reviewer #2 do reduce the novelty of your claims somewhat, increasing the importance of demonstrating the translational potential of your findings more directly, as requested by reviewer #4, who recommends that you assess the effect of light on a mouse burn wound model.

In light of the reviews (below), we will not be able to accept the current version of the manuscript, but we would welcome resubmission of a much-revised version that takes into account the reviewers' comments. We cannot make any decision about publication until we have seen the revised manuscript and your response to the reviewers' comments. Your revised manuscript is also likely to be sent for further evaluation by the reviewers.

Your revisions should address the specific points made by each reviewer. Please submit a file detailing your responses to the editorial requests and a point-by-point response to all of the reviewers' comments that indicates the changes you have made to the manuscript. In addition to a clean copy of the manuscript, please upload a 'track-changes' version of your manuscript that specifies the edits made. This should be uploaded as a "Related" file type. You should also cite any additional relevant literature that has been published since the original submission and mention any additional citations in your response. 

Before you revise your manuscript, please review the following PLOS policy and formatting requirements checklist PDF: http://journals.plos.org/plosbiology/s/file?id=9411/plos-biology-formatting-checklist.pdf. It is helpful if you format your revision according to our requirements - should your paper subsequently be accepted, this will save time at the acceptance stage.

Please note that as a condition of publication PLOS' data policy (http://journals.plos.org/plosbiology/s/data-availability) requires that you make available all data used to draw the conclusions arrived at in your manuscript. If you have not already done so, you must include any data used in your manuscript either in appropriate repositories, within the body of the manuscript, or as supporting information (N.B. this includes any numerical values that were used to generate graphs, histograms etc.). For an example see here: http://www.plosbiology.org/article/info%3Adoi%2F10.1371%2Fjournal.pbio.1001908#s5.

For manuscripts submitted on or after 1st July 2019, we require the original, uncropped and minimally adjusted images supporting all blot and gel results reported in an article's figures or Supporting Information files. We will require these files before a manuscript can be accepted so please prepare them now, if you have not already uploaded them. Please carefully read our guidelines for how to prepare and upload this data: https://journals.plos.org/plosbiology/s/figures#loc-blot-and-gel-reporting-requirements.

Upon resubmission, the editors will assess your revision and if the editors and Academic Editor feel that the revised manuscript remains appropriate for the journal, we will send the manuscript for re-review. We aim to consult the same Academic Editor and reviewers for revised manuscripts but may consult others if needed.

We expect to receive your revised manuscript within two months. Please email us (plosbiology@plos.org) to discuss this if you have any questions or concerns, or would like to request an extension. At this stage, your manuscript remains formally under active consideration at our journal; please notify us by email if you do not wish to submit a revision and instead wish to pursue publication elsewhere, so that we may end consideration of the manuscript at PLOS Biology.

When you are ready to submit a revised version of your manuscript, please go to https://www.editorialmanager.com/pbiology/ and log in as an Author. Click the link labelled 'Submissions Needing Revision' where you will find your submission record. 

Sincerely,

Roli Roberts

Senior Editor

PLOS Biology

REVIEWERS' COMMENTS:

Reviewer #1:

General Comments

In a masterful random transposon search, these investigators discovered that the transmembrane histidine kinase, KinB, which is in an operon with an alginate 2-component regulator, AlgB, affects biofilm formation in Pseudomonas aeruginosa. Their findings show that KinB is a dual kinase/phosphatase, but mostly a phosphatase during their in vivo conditions. Through another mutant search, they find that BphP-AlgB-KinB forms a “three-component” system where AlgB is activated by the kinase activity of light-sensor BphP and inhibited by the phosphatase activity of KinB. Furthermore, they demonstrate that P. aeruginosa biofilm formation can be modulated simply by tuning the intensity of far-red light in which the strain is grown. This is amazing that Pseudomonas virulence phenotypes are under photo control, and that the molecular genetic mechanisms can be explained. 

Specific Comments

1. Line 107 says WT PA14 exhibits a rugose-center/smooth-periphery colony biofilm, and the ∆rhlR mutant forms a larger hyper-rugose biofilm. Line 109 says RhlR impedes biofilm formation, and yet the wild-type (RhlR+) is said to make a “biofilm.” This is confusing, so a better definition of biofilm is needed instead of calling everything a biofilm. By definition, does WT PA14 (RhlR+) form a biofilm even though it is mostly smooth? There appears to be 3 phenotypes on the Fig 2A plates: WT semi rugose biofilm, hyper-rugose biofilm and smooth biofilm. Line 123 says kinB “mutants failed to form biofilms…” Should this say “mutants formed smooth biofilms” to be consistent? Should line 128 say “KinB is essential for ‘hyper-rugose’ biofilm formation”? Why is smooth less of a biofilm? A short description of how the Congo red agar phenotypes compare with other types of Pa biofilms (e.g., static ring, flow cell) would be helpful.

2. Line 117 – (Fazli et al., 2014) was not added to the References list and should have a # in the text.

3. Line 139 says KinB is a HK that transfers phosphate to AlgB (ref. 19), but the Fig. 1 model contradicts this and shows KinB + ?stimulus as a phosphatase that only dephosphorylates AlgB. This is confusing, so maybe the legend should state it also phosphorylates AlgB in the presence of an unknown KinB stimulus.

4. Fig. 2A has a lot of genotypes/phenotypes to sort through. The conclusion at Line 147 says AlgB represses (WT, semi-rugose) biofilm development. But WT PA14 is algB+ and has a WT semi-rugose biofilm phenotype [as did an algB(stop) mutant, which should be smooth according to line 147]. The kinB mutant has a smooth biofilm, so it’s fair to say KinB activates the WT biofilm phenotype. It is appropriate to look at AlgB, the logical partner of KinB, and a kinB algB mutant goes back to a WT phenotype; that fits. I’m not sure that algB overexpression, producing a smooth phenotype (Fig 2A-12) is a reliable phenotype. Perhaps Line 147 should say that “these data suggest that …”

5. Line 160 invokes an unknown negative regulator of biofilms (controlled by AlgB) based on strains with algB overexpressed vs. algB(D59N) unphosphorylated. In the WT (Fig 2A-1), the strain is chromosomally AlgB+ but shows the WT semi-rugose biofilm. Perhaps WT with algB overexpressed (Fig 2A-12) has too much AlgB~P, causing an artifact? It may require having algB(D59N) in single copy in the chromosome to really confirm the conclusions.

6. Line 170. This is amazing that AlgB~P is detected in a kinB mutant, but not in WT, suggesting (line 180) that some His-kinase other than KinB phosphorylates AlgB! 

7. The discovery of the bphP gene as encoding the kinase of AlgB was excellent genetics. Much of the manuscript is on the effect of light on phenotypes, which is well done. Perhaps the lesson is that we can’t incubate our plates in just dark incubators anymore as this may mask other phenotypes.

Reviewer #2:

This manuscript by Mukherjee and colleagues describes what they term a “three component system” that is comprised of the known P. aeruginosa regulators, KinB, AlgB, and BphP. The authors provide genetic and biochemical evidence that the KinB kinase dephosphorylates AlgB while the bacteriophytochome kinase, BphP, phosphorylates AlgB. These three proteins coordinately control biofilm formation and the expression of virulence genes. The discoveries outlined in the manuscript provide new understanding of genes that function downstream of the quorum regulator RhlR to control virulence and biofilm gene expression. Moreover, the authors uncover a role for light in the regulation of these genes through the known red-light sensor, BphP.

This study will be of interest to a broad range of scientists, particularly investigators studying bacterial pathogenesis and photobiology. The study is well written and the data are easy to follow. The authors should consider the suggestions below before publication.

1) BphP has been previously described as a quorum regulated gene by Schuster et al (J. Bact 2003, 185:2066) and by Barkovits et al (Microbiology 2011, https://doi.org/10.1099/mic.0.049007-0) so there was some indication that this gene would play a role in some quorum pathway. It’s likely worth touching on this in either the introduction or discussion.

2) Along this same line, BphP has also been shown to be regulated by RpoS (Barkovits 2011). Therefore, it is likely that multiple pathways impinge on the BphP-AlgB axis. This is likely a minor discussion point should the authors choose to address it.

3) The idea that light, via BphP, may be playing a role in infection/parthogenesis is interesting. However, the Hung group has shown that a BphP mutant is not attenuated in a vertebrate model (Chand et al, J. Bact 2011). Clearly, there is more to do on the infection with respect to light and other treatments, but this result from the Hung group could be mentioned/discussed. 

4) I may have missed it, but does apo BphP have the same activity and response regulator specificity?

5) It is clear from the authors’ data that red-light has the most potent effect of BphP-dependent regulation of gene expression. However, the result that blue, red, and far-red light all activate BphP goes against what I understand about the energy dependence of the bacteriophytochromes. This is a very interesting piece of data that is worth discussing more. In particular, I think it would be useful to revisit some the bacteriophytochome literature to see if there are other examples of these proteins that are activated by blue light. Is there any indication from the published Pa-BphP absorption spectra that this protein can be activated by blue light?

6) A handful of references (e.g. 23 & 29, others) are missing the journal name.

Reviewer #3:

[identifies himself as Carl E Bauer]

This manuscript from the Bassler laboratory effectively and convincingly demonstrates that P. aeruginosa has a very interesting complex phosphoryl relay that controls both biofilm formation and virulence factor production in response to light exposure. Their study identifies all of the major components of this signaling event and uses a strong and convincing combination of in vivo genetics and in vitro biochemistry to show roles of these various signaling components. My overall conclusion is that this manuscript is a real tour de force! It’s a stunningly good example of a complete body of work that will likely open up a whole new avenue of research on the role of photoreceptors in the control of biofilm formation and virulence. 

I only have a few minor suggestions.

Line 67-69: B12 is also recently recognized as a light absorbing chromophore for a photoreceptor that controls photosynthesis gene expression. B12 thus represents a new member of the photoreceptor family. See: Trends in Biochemical Sciences. 41:647-50; Elife. 2018 Oct 3;7. pii: e39028. doi: 10.7554/eLife.39028. 

Figure 1: Is there any evidence that BphB is membrane associated? As far as I know, bacterial phytochromes are not membrane bound. Also, this figure may be oversimplified. As shown in lines 271-273, KinB is capable of autophosphorylation and transferring a phosphate to AlgB in vitro. The authors rightfully conclude that kinB may be providing a phosphate to AlgB in vivo but the conditions for this need to be worked out. The likely dual role of KinB phosphorylation and dephosphorylation of AlgB is not reflected by this figure.

Reviewer #4:

In this manuscript, the authors demonstrate that BphP is a SK that can phosphorylate the RR AlgB. AlgB has been known for some time to work with its cotranscribed SK KinB, however KinB appears to function primarily as a phosphatase rather than a kinase. This manuscript solves the puzzle of what else might be phosphorylating AlgB, by identifying the far red light sensing SK BphP.Following phosphorylation assays, the methods employed include basic in vitro assays of phenotype and biofilm formation. The manuscript is convincing as far as it goes, although for a journal like PLoS Biology I would expect to see more of the medical relevance of BphP’s role explored e.g. employing an animal infection model to assess virulence of BphP mutants vs WT, and also testing the role of light as a therapy.

Other comments:

Line 38 and elsewhere: “three-component” system – please rephrase. This is still a two-component system (i.e. a system with sensor kinase(s) and response regulator(s)). This two-component system happens to have two sensor kinases (KinB and BphP) and one response regulator (AlgB). One of those SKs is primarily a phosphatase (KinB) while the other is primarily a kinase (BphP).

Line 39: “We propose that KinB-AlgB-BphP constitutes a “three-component” system with AlgB acting as the node at which varied sensory information is integrated. This network architecture provides a mechanism enabling the bacteria to integrate at least two different sensory inputs, quorum sensing and light, into the control of collective behaviors”. Would be helpful to clarify that the link to quorum sensing is coming in simply at the transcriptional level (i.e. RhlR activating transcription of AlgB and KinB) rather than as a consequence of AlgB being controlled by the sensor kinases KinB & BphP.

Line 87: “We demonstrate that P. aeruginosa biofilm formation and virulence gene expression are repressed by far-red light”. I am puzzled as to why no images/data are shown for the single bphPstop mutant in figure 4A, C & D. Could these be added? These would show that it was bphP that was acting as the far-red light sensor. To some extent this information is in Figure 5, but it would be neater to have in figure 4 too.

Table S1: To help the reader please add PAO1 gene names to this table.

Line 139-147. The AlgBSTOP mutant is an unusual choice of mutant. AlgB and KinB are encoded together in an operon, with algB being the upstream gene; there is a danger that a premature stop codon within algB would significant impact kinB expression. 

Line 154: Assumption that D59N mutant of algB is not phosphorylatable. Cite Fig S4 here as this isn’t always the case eg. E. coli CheYD57N is phosphorylated at serine 56 instead (Appleby and Bourret, 1999). 

Figure S3: Legend mentions panels A-E but only A-C are present.

In vitro phosphorylation assays: Nice data. Line 614 - Please specify the actual protein concentrations used for these (it is not clear enough to say equimolar). 

Figure 3 and elsewhere: Cognate needs to be defined carefully and explicitly. There are two different versions of it in the literature: i) it can be used to mean transcribed from the same place e.g. within an operon or ii) the SK that recognises and phosphorylates the RR. 

Figure 3: Agreed that BphP is “a” cognate HK for AlgB but not convinced that it is “the” (as in the only) HK for AlgB. What about KinB? Could there be others?

Line467 onwards: Too speculative. How much light would P. aeruginosa be exposed to when infecting humans/mammals? Inside the lungs is pretty dark whether it is day or night. Medical significance is unclear. 

Line486: The idea of testing the effect of light on a burn wound infection is exciting and would massively increase the impact of this paper (if it works). I recommend the authors do these experiments – there are plenty of good mouse burn wound models available.

---

## [Editor Report · Decision Letter 2]

13 Nov 2019

Dear Bonnie,

Thank you for submitting your revised Research Article entitled "Photo sensing and quorum sensing are integrated to control Pseudomonas aeruginosa collective behaviors" for publication in PLOS Biology. The Academic Editor and I have now assessed your revisions and your responses to the reviewers.

Based on this assessment, we're delighted to let you know that we're now editorially satisfied with your manuscript. However before we can formally accept your paper and consider it "in press", we also need to ensure that your article conforms to our guidelines. A member of our team will be in touch shortly with a set of requests. As we can't proceed until these requirements are met, your swift response will help prevent delays to publication. Please also make sure to address the data and other policy-related requests noted at the end of this email.

*Copyediting*

*Published Peer Review History*

*Early Version*

*Submitting Your Revision*

Best wishes,

Roli

Senior Editor

PLOS Biology

---

## [Editor Report · Decision Letter 3]

26 Nov 2019

Dear Dr Bassler,

On behalf of my colleagues and the Academic Editor, Michael T. Laub, I am pleased to inform you that we will be delighted to publish your Research Article in PLOS Biology. 

Early Version

PRESS 

Kind regards,

Sofia Vickers

Senior Publications Assistant

PLOS Biology

On behalf of, 

Roland Roberts,

Senior Editor

PLOS Biology